# GNTD: reconstructing spatial transcriptomes with graph-guided neural tensor decomposition informed by spatial and functional relations

Tianci Song[1], Charles Broadbent[1] & Rui Kuang [1] ✉

Spatially-resolved RNA profiling has now been widely used to understand cells' structural organizations and functional roles in tissues, yet it is challenging to reconstruct the whole spatial transcriptomes due to various inherent technical limitations in tissue section preparation and RNA capture and fixation in the application of the spatial RNA profiling technologies. Here, we introduce a graph-guided neural tensor decomposition (GNTD) model for reconstructing whole spatial transcriptomes in tissues. GNTD employs a hierarchical tensor structure and formulation to explicitly model the high-order spatial gene expression data with a hierarchical nonlinear decomposition in a three-layer neural network, enhanced by spatial relations among the capture spots and gene functional relations for accurate reconstruction from highly sparse spatial profiling data. Extensive experiments on 22 Visium spatial transcriptomics datasets and 3 high-resolution Stereo-seq datasets as well as simulation data demonstrate that GNTD consistently improves the imputation accuracy in cross-validations driven by nonlinear tensor decomposition and incorporation of spatial and functional information, and confirm that the imputed spatial transcriptomes provide a more complete gene expression landscape for downstream analyses of cell/spot clustering for tissue segmentation, and spatial gene expression clustering and visualizations.

Many different types of cells are structurally organized to play distinct and cooperative functional roles in biological tissues. To understand cells' organizations and their functions in tissues, spatial transcriptomics technologies have now been widely used to profile spatially resolved RNA expressions. These spatial transcriptomics technologies both profile gene expressions and retain their spatial localization information in the tissue. In-situ hybridization (ISH) methods use fluorescently labeled probes hybridized to targeted RNA transcripts to measure and visualize gene expression at subcellular resolution, which has evolved from earlier low-gene-throughput single-molecular FISH (smFISH)[1] to high-gene-throughput and even

nearly transcriptome-wide multiplexed error robust FISH (MERFISH)[2] and sequential FISH (seqFISH and seqFISH+)[3–5]. More recently developed in situ capturing (ISC) methods perform RNA sequencing of the whole transcriptome with positional barcodes in a spatial genomic array aligned to locations on the tissue without relying on predefining probes and selecting target genes. These methods range from lower resolution Spatial Transcriptomics (ST)[6] (commercialized as 10x Genomics Visium[7]), to higher resolution Slide-seq[8], or even sub-cellular resolution technologies such as high-definition spatial transcriptomics (HDST)[9] and Spatio-temporal enhanced resolution omics-sequencing (Stereo-seq)[10].

[1]Department of Computer Science and Engineering, University of Minnesota Twin Cities, Minneapolis 55414 MN, USA. ✉e-mail: kuang@umn.edu

While in situ capturing technologies aim to capture and sequence all the RNAs in the whole transcriptome in all the spots on the spatial genomic array, there are still significant limitations. First, in situ capturing has a low RNA capture efficiency ranging from 6.9% with ST (slightly higher with Visium arrays) to as low as 1.3% with Slide-seq and 0.3% with HDST[11]. While the newer high-resolution technologies such as Stereo-seq[10] are improving the capture efficiency, the aggregated signals by RNA read counts can still be as sparse as Visium data in the experiments on real tissues. Furthermore, sample preparation requires highly specific handling of tissue sections and treatments. RNA fixation and permeabilization might fail in some tissue regions due to various possible issues in preparing tissue sections and the array[12]. Thus, reconstructing the whole spatial transcriptomes from the incomplete RNA profiling due to these inherent limitations of the spatial technologies is often a necessary step for many critical downstream analyses such as clustering spatial spots for tissue segmentation, detecting spatially co-expressed gene modules, and enhancing expression of spatially variable genes.

In this research work, we introduce a graph-guided neural tensor decomposition (GNTD) model for reconstructing whole spatial transcriptomes in the tissue by integrating spatial relations among the capture spots and the functional relations among the genes. GNTD is a 3-layer neural network designed to model the completion of a three-way tensor in spatial coordinates ($x$ and $y$ modes) and gene ($g$-mode) with hierarchically structured components. GNTD learns nonlinear relations among all the elements in each mode for constructing the factors of canonical polyadic decomposition (CPD) of the tensor. To overcome the overfitting issue in sparse tensors, a graph regularization is also introduced to smooth the imputation by spatial information among the spots in the array and functional relations among the genes in the Protein-Protein Interaction (PPI) Network. The graph regularization is based on the prior knowledge that neighboring spots often share similar gene expressions and functionally related genes are more likely co-expressed.

GNTD is a hierarchical nonlinear tensor decomposition model based on graph-guided neural training. First, GNTD architecture models latent features at different levels such that the hierarchical representations can capture the more complex nature of the tensor data. Second, GNTD is regularized with a Cartesian product graph, which imposes structural relations to avoid overfitting for learning the hierarchical representations in the neural tensor decomposition. GNTD is a different method designed for spatial transcriptomics data imputation and analysis, compared with those imputation methods for single-cell gene expressions. First, the spatial gene expression data are naturally manifested in a high-order structure with gene expressions measured in 2D or 3D locations. The high-order structure implies more complex relations among the spatial coordinates and the genes as opposed to simple sample-gene relations. Second, the spatial arrangement of the spots suggests functional continuity in the tissue vicinity such as similar cell types or correlated (marker) gene expressions, which requires explicit spatial modeling. Finally, the imputation of highly sparsely expressed genes can often benefit from other functionally related genes.

Modeling spatial dependency is critical in spatial transcriptomics data analysis. For example, conditional autoregressive prior (similar to the Laplacian of the spatial graph) has been used in generalized linear models with zero-inflated Poisson link function[13], and FIST[12] used a Cartesian product graph to reduce the complexity of a joint representation of two spatial chain graphs and PPI network to incorporate both spatial and functional dependence. GNTD employs a similar Cartesian product graph between a spatial graph and PPI network with hierarchical CPD rather than the standard CPD as FIST[12]. While FIST is a gradient descent algorithm based on multiplicative updates for standard CPD with product graph regularization, GNTD is a back-propagation training algorithm to learn a non-linear hierarchical CPD in a neural network with product graph regularization. Thus, GNTD is a more advanced method than FIST by generalization to hierarchical and nonlinear tensor decomposition based on neural network training.

The imputation task in this study focuses on modeling and estimating the missing expressions over the measured spots, which is similar to imputing dropouts in scRNAseq data[14]. This task is different from several other imputation or imputation-related tasks in broader or other contexts. For example, spatial deconvolution methods map scRNAseq profiles onto the spatial locations[15], and some other methods impute gene expressions in the unmeasured locations for higher resolution and/or better coverage[16,17]. There are also methods for estimating the expressions of unprofiled genes based on probed genes in in situ hybridization data. We will discuss the relation to these other different tasks in "Discussion" and the supplementary document.

## Results

### Overview of GNTD

The architecture of GNTD is shown in Fig. 1. GNTD models the observed expression profile of spatial transcriptomics as a three-way tensor in spatial coordinates ($x$ and $y$ modes) and genes ($g$-mode). GNTD learns nonlinear latent factors representing each mode in the tensor and reconstructs the tensor with these factors through a 3-layer neural network composed of a hierarchy of linear embedding, nonlinear mapping, and nonlinear aggregation layers. The nonlinear layers explore nonlinear interaction within and across the latent factors in all the modes to characterize more complex underlying nonlinear structures, and thus this hierarchical structure is beyond simple multilinear structure assumed by conventional tensor decomposition methods.

The first layer learns an underlying linear embedding in each mode representing the linear factors. The second layer introduces nonlinear mappings among all the linear factors within each mode with nonlinear activations. Finally, the last layer aggregates nonlinear factors along each mode and structures the loss function of the neural network as CPD regularized by the graph Laplacian of the product graph of the spatial graph and the PPI network. The hierarchical representations can capture latent features at different levels of abstraction of data with complex patterns such as highly irregular and nonconvex shapes of the tissue regions in spatial transcriptomics data. Such hierarchical models have been shown useful in semi-nonnegative matrix factorization in face recognition, topic modeling in text analysis, and other research problems[18,19]. To better infer the unobserved expression profile with the learned nonlinear latent factors, GNTD also leverages the prior knowledge of spot spatial arrangement and gene functional modules encoded in the spatial neighborhood and protein-protein interaction (PPI) graphs. GNTD combines these graphs via Cartesian product and applies the graph Laplacian regularization to impose spatial and functional similarity over nonlinear latent factors such that the observed and unobserved entries in the reconstructed tensor tend to share similar expressions if they are spatially adjacent or functionally proximate. The detailed definition of GNTD neural network and the optimization algorithm are given in "Methods".

### GNTD imputes spatial gene expressions more accurately in in-silico simulations

We conducted simulations to compare GNTD and the existing tensor decomposition models for imputing spatial transcriptomics data. The comparison includes two nonlinear tensor decomposition models, CoSTCo[20] and DTD[21,22], as well as one graph-regularized tensor decomposition model FIST[12], as reviewed in "Compared methods". We first constructed a simulated spatial transcriptomics dataset with the same spatial layout in DLPFC 151673 section, where the simulated data was generated over six cortical layers and white matter (WM) and manually segmented by the annotation in the original study as shown in Fig. 2a.

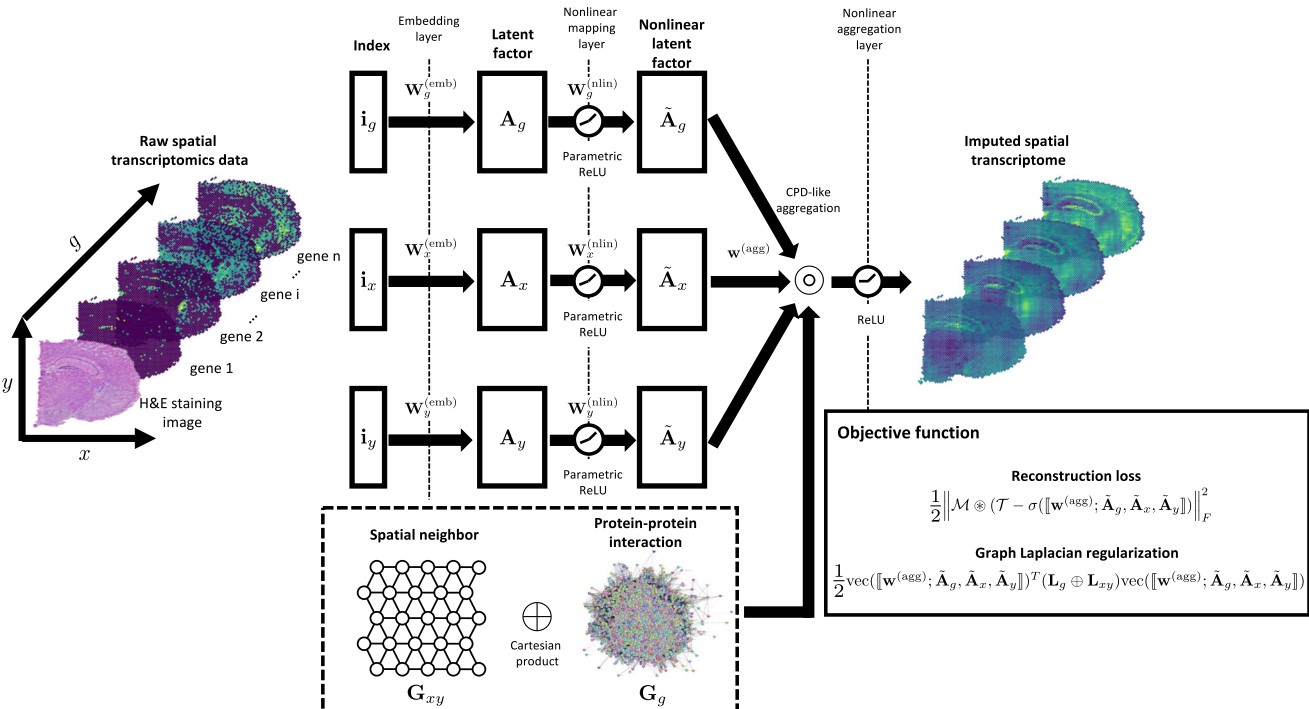

**Fig. 1 | The three-layer neural network architecture of GNTD.** Spatial gene expressions are modeled by a 3-way tensor by spatial coordinates ($x$ and $y$ modes) and genes ($g$-mode). The first layer learns the weights $\{\mathbf{W}_x^{(\text{emb})}, \mathbf{W}_y^{(\text{emb})}, \mathbf{W}_g^{(\text{emb})}\}$ representing the linear factors $\{\mathbf{A}_x, \mathbf{A}_y, \mathbf{A}_g\}$ in each mode by a linear embedding. The second layer introduces nonlinear mappings among all the linear factors within each mode by nonlinear activations over the weights $\{\mathbf{W}_x^{(\text{nlin})}, \mathbf{W}_y^{(\text{nlin})}, \mathbf{W}_g^{(\text{nlin})}\}$ to learn the nonlinear factors $\{\tilde{\mathbf{A}}_x, \tilde{\mathbf{A}}_y, \tilde{\mathbf{A}}_g\}$. In the last layer, the loss function of the neural network is aggregated by learnable coefficients $\mathbf{w}^{(\text{agg})}$ and structured as a weighted CPD regularized by the graph Laplacian of the product graph of the spatial graph $\mathbf{G}_{xy}$ and the PPI network $\mathbf{G}_g$. Once the GNTD model is trained, the indexes $\{\mathbf{i}_x, \mathbf{i}_y, \mathbf{i}_z\}$ of the tensor entries can be used as inputs to query the reconstructed values.

We then simulated the expressions of 50 spatially variable genes by sampling UMI counts from two different negative binomial (NB) distributions. The first distribution was generated with a random number of successes $r$ in the range of [10, 100] and probability of success $p = 0.85$ in a single trial for some randomly selected highly expressed regions. For the remaining lowly expressed regions, we used another NB distribution with $r/2$ successes and probability of success $p = 0.95$. In addition, we also simulated 50 ubiquitously expressed genes by sampling UMI counts from a background NB distribution with random $r \in [5, 50]$ and $p = 0.85$ over all the regions. Zero inflation is then introduced by setting a certain percentage (40% or 80%) of entries to zeros in the sampled data. Note that the density of the simulation data with 40% zero inflation is around 50%, which is close to the density of ISH data generated by seqFISH[5]. And, the density of the simulation data with 80% zero-inflation is 13%, which is close to the density of sparser data from Visium (See Supplementary Table S1). For the simplicity of this simulation, we set the PPI network to be a diagonal graph without functional information among the genes.

First, we evaluated the performance of detecting the spatial domains by clustering the spots in the raw data and its imputation generated by GNTD and the baseline models on the simulated data with zero-inflation rates 40% and 80%. The results are shown in Fig. 2b under different CPD ranks and two choices of the graph regularization weight $\lambda = 0.01$ or 0 (no regularization). It is evident that clustering on the data imputed by GNTD consistently outperforms the clustering on the raw data and its imputation by the other tensor-based models on the simulated data with both low and high zero inflation rates. All the tensor-based models provide better imputation for spot clustering than the raw data at all compared ranks when the zero inflation rate is high at 80%. When the rank is sufficiently large, this is also true in the lower zero inflation rate of 40%. The superior performance of GNTD ($\lambda = 0.1$) and FIST ($\lambda = 0.01$) to GNTD ($\lambda = 0$) and FIST ($\lambda = 0$) also

confirms that the spatial localization encoded in the graph regularization is playing an essential role in the imputation. The visualization of the simulation with a high zero inflation rate (80%) in Fig. 2c shows that GNTD imputation accurately identifies all tissues regions while raw data and other imputed data fail to delineate tissue region borders. It is not surprising that CoSTCo and DTD detect spatial domains with less spatial continuity since these models do not incorporate spatial relations among the spots. Moreover, GNTD ($\lambda = 0$) also performed worse than GNTD ($\lambda = 0.1$), especially in the simulated data with a high zero inflation rate (80%), and missed one delicate tissue region in simulated data, implying spatial proximity could improve spatial domain detection when simulated data is more sparse and noisy.

Next, we evaluated the performance of gene spatial pattern recovery by the raw data and the imputed data generated by GNTD and the baseline models by calculating the AUC scores over the ranking of the spots by their imputed expressions, where the spots are labeled as either highly expressed or lowly expressed in the ground truth of each spatially variable gene. The results are shown in Fig. 2d. It is clear that all 50 spatially variable genes from the imputation generated by GNTD have AUC scores greater than 0.95, while less than 50% of the spatially variable genes have the AUC scores at the same level in the imputation by CoSTCo and DTD. Interestingly, around 80% spatially variable genes have AUC scores greater than 0.95 by the imputation by FIST. In addition, gene spatial patterns recovered by GNTD match well with the ground truth patterns compared to the imputation by the other models (Fig. 2e and Supplementary Fig. S3). GNTD ($\lambda = 0$) without graph regularization also exhibits good performance, but the imputed expressions within the same region are less consistent, which further indicates that the spatial proximity in the graph indeed contributes to refining the spatial expression patterns.

In Supplementary Fig. S1, we also compared GNTD with the three autoencoder (AE)-based models using the data reconstructed from the

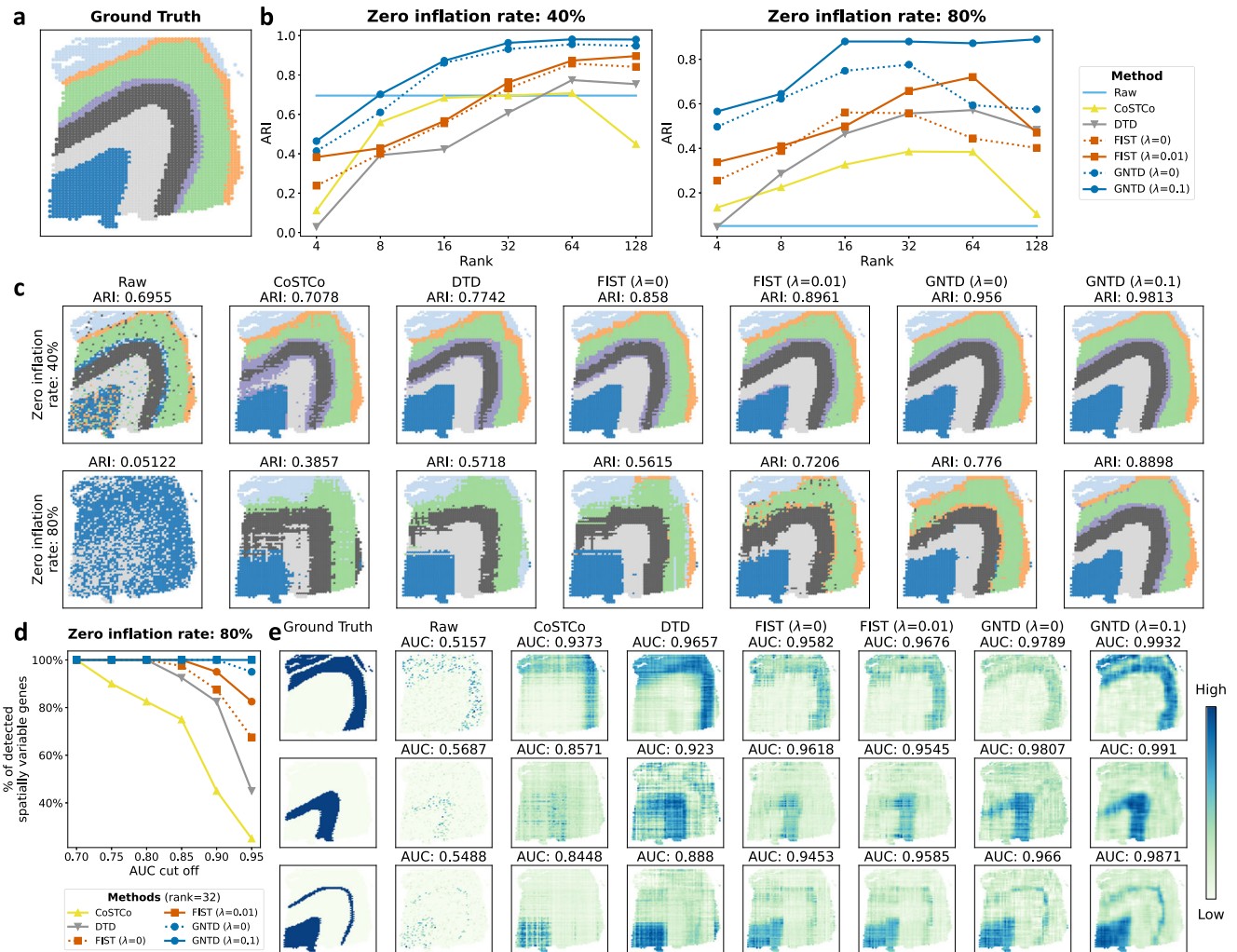

**Fig. 2 | Spatial domain detection and gene spatial pattern recovery in simulated spatial transcriptomics data. a** Ground-truth segmentation of 6 cortical layers and white matter (WM) for simulated spatial transcriptomics data based on the annotation of the human dorsolateral prefrontal cortex (DLPFC) section 151673. **b** Spot clustering performance on the raw data and the imputed data by CoSTCo, DTD, FIST ($\lambda = 0$ or 0.01), and GNTD ($\lambda = 0$ or 0.1) at different ranks in the simulated spatial transcriptomics data with 40% or 80% zero inflation rate. **c** Visualization of the spatial domains detected by spot clustering on the raw data and the imputed data of the simulated spatial transcriptomics data with 40% and 80% zero inflation rates. The imputed data with the best rank by each tensor decomposition method

was used in the visualization. **d** Spatially variable genes detection comparison. The plot shows the percentage of correctly detected spatially variable genes by the AUC thresholds of the recovered highly expressed spots in the more sparse simulated spatial transcriptomics data with 80% zero inflation rate. **e** Spatial patterns visualization of three example genes by their expression in the ground-truth data, raw data, and the imputation data of the simulated spatial transcriptomics data with 80% zero inflation rate. Note that in (**d**) and (**e**), a higher AUC indicates a better consistency between the imputed or raw expressions and the ground-truth expression over the spots for the gene. Source data for (**b**) and (**d**) are provided as a Source Data file.

AE embedding on the simulation data, SEDR[23], STAGATE[24], and GraphST[25] (See "Compared methods"). Note that SEDR, STAGATE, and GraphST also utilize spatial relations with graph convolution. In this comparison, two different kinds of loss were used for training these AE-based models. In the first setting, the loss of only non-zero entries was used for training these AE-based models the same as for training GNTD. In the second setting, the loss of all entries (both zero and non-zero entries) was used to train the AE-based models as they were trained in the original studies for embedding. Based on the ARIs and AUCs in Supplementary Fig. S1, it is evident that GNTD outperforms the AE-based models in both spatial domain detection and gene spatial pattern recovery by a large margin. While the AE-based models perform relatively better on the low zero-inflation data (40%) by revealing some spatial patterns, their imputation on the high zero-inflation data (80%) show no or much less spatial content. Training with all entries did improve the imputation by STAGATE but not consistently for SEDR and GraphST.

To further investigate if the imputation could introduce false negative or false positive spatially variable genes, we applied SPARK[26] to detect spatially variable genes in the imputation of the simulation data with a high zero-inflation rate (80%). Notably, the results in Supplementary Fig. S2 show that GNTD did not introduce any false positive or false negative spatially variable genes in the detection while all other methods introduced a significant number of either false positive or false negative spatially variable genes, or even both. The imputed expressions of each spatially variable gene are also fully visualized in Supplementary Fig. S3.

## GNTD imputes significantly more accurate spatial gene expressions in Visium data

To evaluate the imputation performance, we also applied GNTD, the three tensor-based models (CoSTCo, DTD, and FIST), and the three AE-based models (SEDR, STAGATE, and GraphST) to perform 10-fold cross-validation on all the 22 Visium spatial transcriptomics datasets.

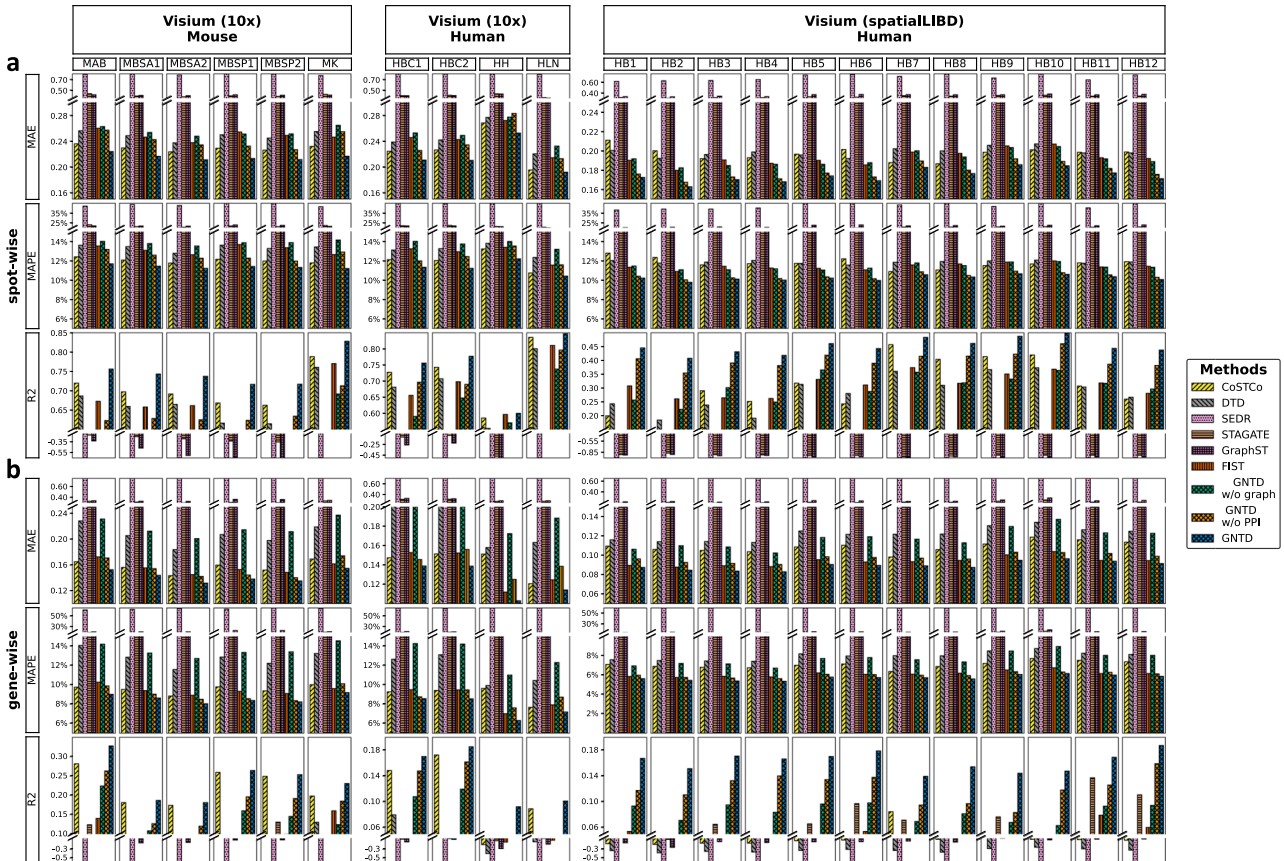

**Fig. 3 | Evaluation of imputation accuracy on 22 Visium datasets.** The evaluations are measured by MAE, MAPE and $R^2$ among 3 AE-based deep learning models SEDR, STAGATE, and GraphST, as well as 4 tensor-based models CoSTCo, DTD, FIST, and GNTD. Each bar shows the mean of the imputation performance over all the spots or all the genes. **a** Spot-wise 10-fold cross-validation and **b** Gene-wise 10-fold cross-validation. Source data for (**a**) and (**b**) are provided as a Source Data file.

To better understand the results, we also added GNTD without any graph regularization (GNTD w/o graph) and GNTD with spatial graph regularization but no PPI (GNTD w/o PPI) as baseline models. We measured the cross-validation performance for all the models in both spot-wise and gene-wise cross-validations with 3 metrics, MAE, MAPE, and $R^2$, where the detailed design of spot-wise and gene-wise cross-validation and the definitions of the evaluation metrics are given in "Imputation evaluation by cross-validation" and "Evaluation metrics" respectively.

GNTD consistently achieved the best spot-wise and gene-wise imputation with the lowest MAE and MAPE, and the highest $R^2$ as shown by the comparisons in Fig. 3. Nonlinear tensor-based models CoSTCo and DTD exhibit worse spot-wise and gene-wise imputation performance than GNTD without graph regularization ($\lambda = 0$), which further indicates that the hierarchical representation by linear and nonlinear factors could better model complex interactions among the genes and spatial locations in the spatial transcriptomics data. The observation that GNTD also outperforms FIST suggests that non-linearity within factors indeed improves the imputation in both the accuracy and the correlation of spatial expressions. In addition, GNTD also shows better evaluation performance in both spot-wise and gene-wise imputation than its variants, GNTD w/o graph and GNTD w/o PPI. This result suggests the importance of the functional relations among genes as well as spatial relations among spots in the spatial transcriptomics data imputation. Note that the $R^2$ metric as defined in Eq. (11) can be negative when the overall prediction is worse than the mean. This can happen very often in highly sparse data if the non-zero entries are not correctly predicted from the majority of zeros. The three AE-based models performed poorly in both spot-wise and gene-

wise imputation evaluation since they are specifically designed and trained for learning the latent representation and might suffer from overfitting of training with non-zero entries in the cross-validation evaluations. Furthermore, we also examined the mean and the variance of MSE to check the robustness of the GNTD imputation on these 22 Visium datasets in the 10-fold cross-validation (Supplementary Fig. S4). The MSEs in both spot-wise and gene-wise experiments are consistent across the 10 folds.

We further analyzed the role of hyper-parameter tuning for GNTD in both the spot-wise and the gene-wise imputation evaluations. We first examined the rank selection for tensor decomposition by the imputation performance by MSE for all the tensor-based models on all the 22 Visium datasets (Fig. 4a and Fig. 5a). It is expected that the MSE of all tensor-based models monotonically decreases as the rank increases within the specific range (rank = {8,16,32,64}) in most Visium datasets since reasonably high ranks generally capture more complex interactions. The best performance of GNTD among all the tensor-based models at all the tested ranks suggests that the nonlinear factors learned by GNTD are more informative in capturing the nonlinearity in the spatial gene expressions. Interestingly, the performance of CoSTCo and FIST degrades at a relatively higher rank (rank = 128) potentially due to over-fitting. This degradation is more significant in sparser datasets. It is also important to note that the results are highly consistent across all 22 datasets, which is strong evidence for generalization to all Visium datasets with the same setting.

Next, we explored the importance of the weight ($\lambda$) on Cartesian product graph regularization in GNTD (Figs. 4b and 5b). In the imputation performance by MSE for GNTD under different weights, we observed that the optimal weight is always either 0.01 or 0.1 in the 22

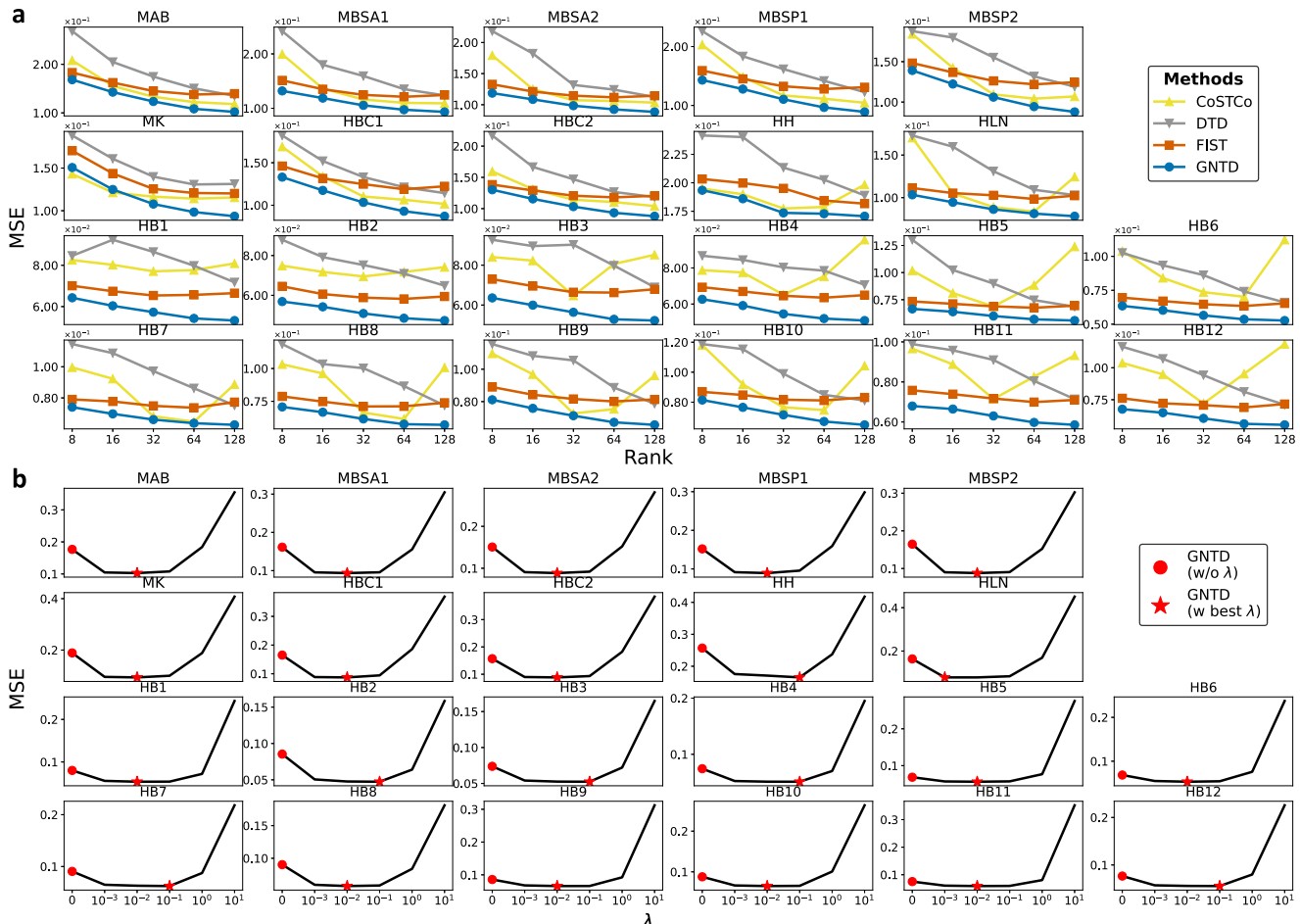

**Fig. 4 | Hyper-parameter tuning on 22 Visium datasets in spot-wise cross-validation. a** Spot-wise 10-fold cross-validation comparison by MSE by varying the rank in tensor decomposition. Each dot shows the mean MSE over all the spots. **b** Spot-wise 10-fold cross-validation MSE of GNTD at the best rank by varying $\lambda$. In the plot, each dot shows the mean MSE over all the spots. Source data for (**a**) and (**b**) are provided as a Source Data file.

Visium datasets. The better performance of GNTD with optimal $\lambda$s than GNTD ($\lambda = 0$) without graph regularization again confirms the important role of graph regularization to guide the imputation by integrating prior information of spatial relations among spots and functional relations among genes encoded in the Cartesian product graph. The declining performance of GNTD after $\lambda > 0.1$ suggests that when too much belief is put on the prior knowledge, the imputation can be corrupted as the prior knowledge of the relations is imperfect.

### GNTD imputation leads to better spatial domain detection in DLPFC sections and human breast cancer sections

To provide more quantitative measures of the quality of the imputed data, we evaluated spot clustering performance by adjusted rand index (ARI) on the raw data and the imputed data in the human dorsolateral prefrontal cortex (DLPFC) sections, based on 6 cortical layers and white matter (WM) manually annotated with morphological features and layer-specific gene markers. GNTD was compared with the tensor-based models, CoSTCo, DTD, and FIST, and the AE-based models (SEDR, STAGATE, and GraphST) in all the 12 DLPFC sections. The results are shown in Fig. 6.

GNTD outperformed all the other models in all the 12 datasets with the overall best ARI in spot clustering with the imputed data using either all genes (median ARI = 0.45) (Fig. 6a) or highly variable genes (median ARI = 0.52) (Fig. 6b). Spot clustering with highly variable genes is generally better than that using all genes by focusing on potential layer-specific marker genes to better define different spatial

domains in the comparison. The spot clustering performance of CoSTCo (median ARI = 0.24 for all genes and median ARI = 0.25 for highly variable genes) and DTD (median ARI = 0.29 for all genes and median ARI = 0.30 for highly variable genes) is worse than the performance of using the raw data, potentially due to the over-expressiveness of nonlinearity even when using highly variable genes.

It is interesting to observe that GNTD also significantly improves the spot clustering performance compared with the two variants, GNTD w/o graph without any graph regularization (median ARI = 0.39 for all genes and median ARI = 0.42 for highly variable genes) and GNTD w/o PPI with spatial graph regularization but no PPI (median ARI = 0.41 for all genes and median ARI = 0.47 for highly variable genes). The observation again emphasizes the importance of both the spatial relations among the spots and functional relations among the genes in imputation used for spatial domain detection.

To show intuitively how GNTD imputation could accurately detect spatial domains, we further examined the spot clustering results using highly variable genes on the DFLPC 151673 section (Fig. 6c). Most of the baseline methods obtained worse ARI than clustering using the original raw data, and the identified spatial domains are either substantially noisy or unable to match the layer patterns. GNTD and its two variants exhibit better ARI than clustering using the raw data. Moreover, GNTD, leveraging both spatial relations among spots and functional relations among genes in the imputation, could delineate continuous spatial domains with smooth boundaries largely agreeing with the layer structures, while the two variants are less accurate. In addition, we also

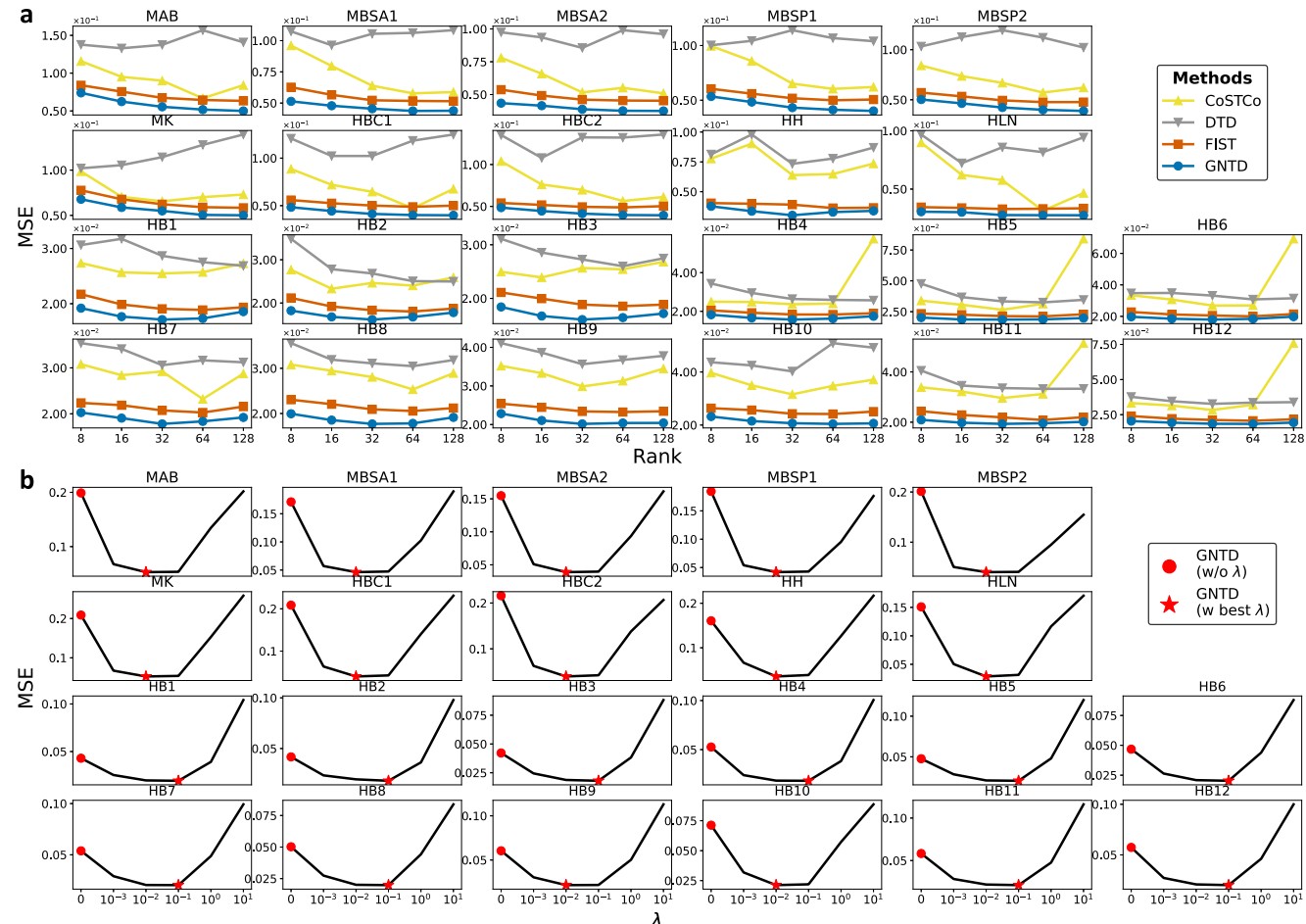

**Fig. 5 | Hyper-parameter tuning on 22 Visium datasets in gene-wise cross-validation. a** Gene-wise 10-fold cross-validation comparison by MSE by varying the rank in tensor decomposition. Each dot shows the mean MSE over all the genes. **b** Gene-wise 10-fold cross-validation MSE of GNTD at the best rank by varying $\lambda$. In the plot, each dot shows the mean MSE over all the genes. Source data for (**a**) and (**b**) are provided as a Source Data file.

applied uniform manifold approximation and projection (UMAP) to map the spots by highly variable genes onto two-dimensional UMAP space in the raw data and the imputed data. We observed that in the clustering of the imputed data by GNTD, the spots from distinct layers are well separated with a spatial trajectory derived from the adjacency in the UMAP space following the chronological order of cortex layer development, whereas in the mappings of the other imputed data, the spots tend to be highly entangled showing inconsistent spatial trajectory that disagrees with the chronological order of cortex layer development. Note that DTD, STAGATE, and GraphST also largely captured most of the chronological order of the layers which is consistent with their relatively higher ARI in clustering.

We next tested the same spot clustering using highly variable genes on the data of the human breast section, which is mixed by four primary tissue types, healthy tissue (Healthy), ductal carcinoma in situ/lobular carcinoma in situ (DCIS/LCIS), invasive ductal carcinoma (IDC), and boundary tissue with low malignancy (Tumor edge) in 20 tissue regions annotated by pathological features. The results are shown in Fig. 6d. Similarly, clustering based on GNTD imputation (ARI = 0.609) shows the best performance over the raw data and the imputed data by the other models. GNTD detects spatial domains that match well with the annotated tissue regions. Interestingly, we also discovered that several seemingly homogeneous spatial regions annotated as DCIS/LCIS or IDC tumor regions are indeed heterogeneous because each of them can be dichotomized into core and surrounding sub-regions highly resembling a tumor region and its microenvironment respectively (Supplementary Fig. S6). This observation is further confirmed

by enrichment analysis on differentially expressed genes between these two sub-regions, where the core sub-region enriches with tumor progression while the surrounding sub-region enriches with tumor-associated immune suppression. The complete enrichment results are shown in Supplementary Table S2.

Similarly, we also projected the raw data and the imputed data of the human breast cancer tissue using highly variable genes onto a two-dimensional UMAP space. There is a clear separation among different tissue regions in the UMAP space computed from the GNTD imputation.

Even if SEDR, STAGATE, and GraphST also consider spatial relations among spots in modeling, the imputation by these three methods provides substantially worse spot clustering performance than their low-dimensional embedding, which might imply that while deep neural network embedding could better characterize spot domains by eliminating noisy and redundant information, the same improvement is not carried over to the reconstructed data from the embedding. All these results corroborate that introducing nonlinearity and incorporating both spatial relations among spots and functional relations among genes enable GNTD to provide informative imputation for spatial domain detection.

## GNTD imputation enhances biological interpretation of spatially co-expressed gene clusters

To demonstrate GNTD imputation can also lead to a better functional interpretation of spatial transcriptomics data, we performed enrichment analysis over spatially co-expressed gene clusters detected from

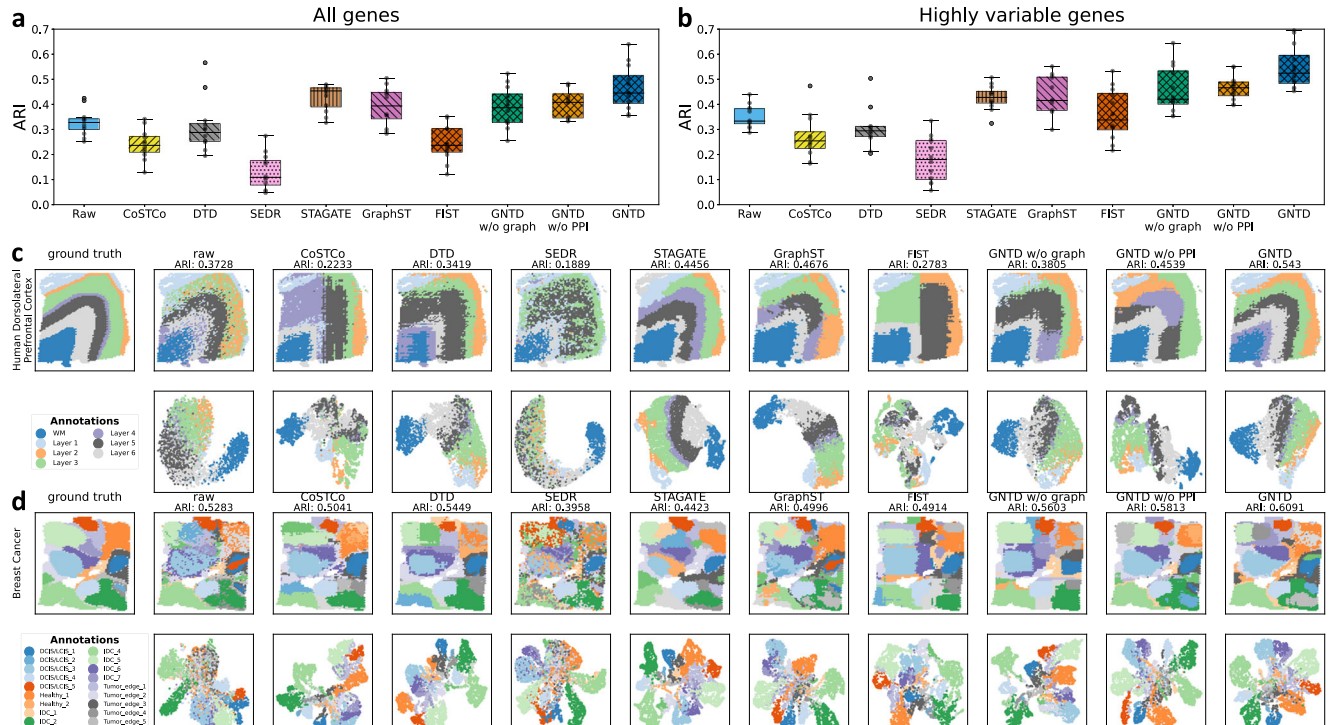

**Fig. 6 | Comparison of detecting layer structures in DLPFC sections and heterogenous tumor tissue regions in human breast cancer section. a** The comparison of spot clustering with all genes in the raw data and the imputed data by tensor-based models, GNTD, FIST, CoSTCo and DTD as well as AE models, SEDR, STAGATE, and GraphST at the best rank on all 12 DLPFC sections. **b** The comparison of spot clustering with only highly variable genes in the raw data and the imputed data by the compared methods at the best rank on all 12 DLPFC sections. In the boxplot in both (**a**) and (**b**), the center line, box limits and whiskers denote the median, upper and lower quartiles and 1.5 × interquartile range, respectively.

**c** Ground-truth of tissue regions and the detected spatial domains by clustering (Upper panel) and the UMAP embeddings of the spots by highly variable genes (Lower panel) of the raw data and the imputed data by the compared methods at the best rank on the DLPFC section 151673. **d** Ground-truth of tissue regions and the detected spatial domains by clustering (Upper panel) and the UMAP embeddings of the spots by highly variable genes (Lower panel) of the raw data and the imputed data by the compared methods at the best rank on the human breast cancer section. Source data for (**a**) and (**b**) are provided as a Source Data file.

the raw data and the imputed data. We measured the average of the log of the minimal q-value of the most significant enriched Gene Ontology (GO) term from each gene cluster on the 22 Visium datasets. Most of the baseline models, except for SEDR, achieved only slightly better enrichment significance in the spatially co-expressed gene clusters discovered by the imputation data compared to those by the raw data. GNTD consistently shows the best enrichment significance over all the gene clusters among all the methods (Fig. 7a). With no surprise, GNTD also performed better than GNTD w/o PPI by incorporating the functional relations among the genes encoded in the PPI network.

We also explored how the rank in the tensor-based models can affect the enrichment significance of the co-expressed gene clusters (Fig. 7b). GNTD achieved the overall best enrichment significance over co-expressed gene clusters by a large margin compared to the other tensor-based models under all the ranks. Generally, the imputation tends to improve the enrichment significance as the rank increases to capture more interactions in factors, suggesting that the inclusion of both nonlinearity and functional relations among genes in the imputation by GNTD achieves more functionally relevant co-expressed gene clusters.

FIST also exhibits considerably better enrichment significance over the spatially co-expressed gene clusters than the other baseline models in relatively sparser datasets (12 DFLPC sections), while CoSTCo and DTD performed better than FIST in relatively denser datasets, which strongly suggests that PPI provides more useful guidance in highly sparse data[12].

Finally, to better understand the co-expressed gene clusters by GNTD imputation on the mouse kidney tissue, we selected 9 co-

expressed clusters characterizing three primary anatomical structures, the cortex, the inner stripe of the outer medulla (ISOM), and the outer stripe of the outer medulla (OSOM), and show the average expression patterns of these co-expressed gene clusters in Supplementary Fig. S7. We further performed enrichment analyses on the co-expressed gene clusters individually and found that the enriched biological processes are highly relevant to their corresponding anatomical structures. The co-expressed gene clusters highly expressed in ISOM enrich nucleotide and ATP metabolisms[27,28], those highly expressed in OSOM enrich catabolic processes of organic and inorganic molecules[29,30], and those highly expressed in cortex enrich the regulation of blood pressure and the transport of cellular metabolites[31,32]. The complete enrichment results are compiled into Supplementary Table S3.

## GNTD performs better imputation on high-resolution spatial transcriptomics data

To further verify the applicability of GNTD to high-resolution spatial transcriptomics data, we repeated all the previous experiments with similar settings on three Stereo-seq datasets including one mouse brain coronal hemibrain section and two mouse olfactory bulb sections, all of which resolve spatial expression at 4 × higher resolution than the Visium data. All three tissue sections are annotated by analyzing differentially expressed genes among clusters from unsupervised graph-based clustering Leiden on the union of spatial and co-expression graphs over spots, and annotations are further validated by comparison with available single-cell data reported for the anatomic regions in the original study[10]. All the results are shown in Fig. 8.

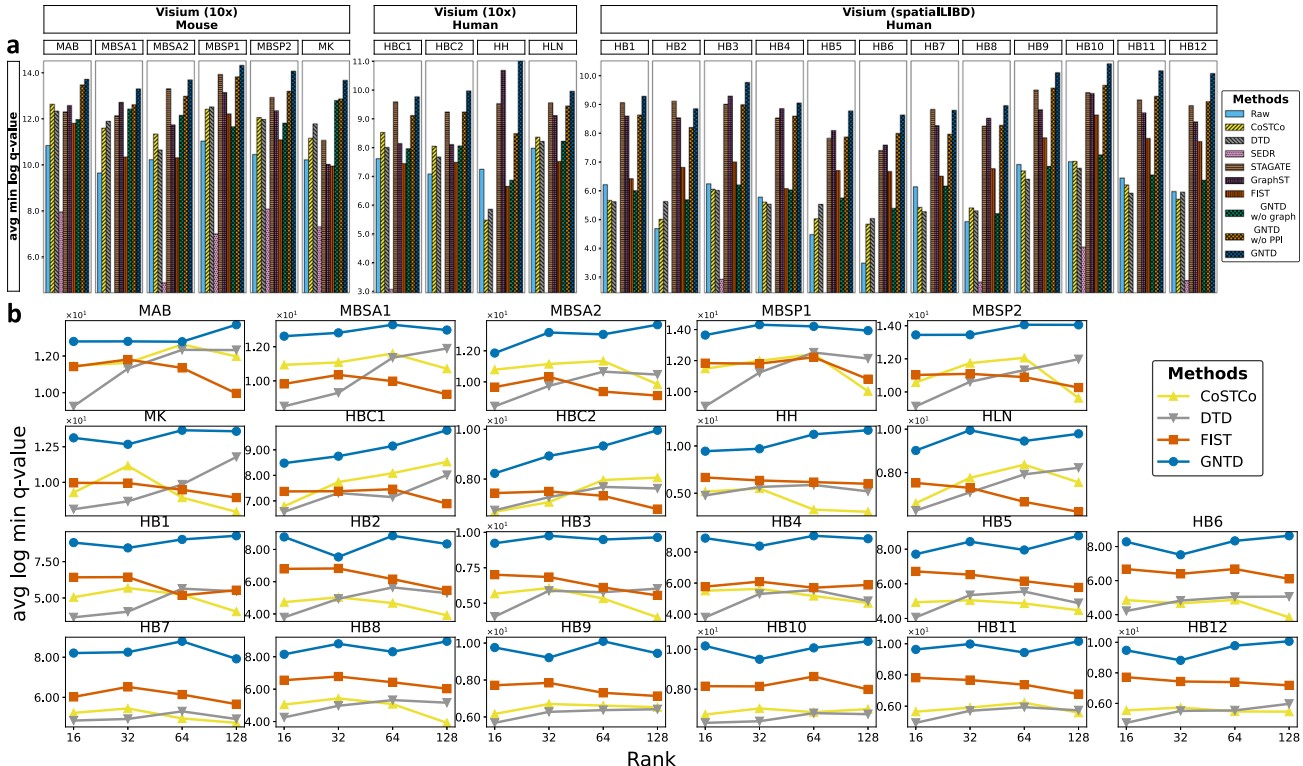

**Fig. 7 | Enrichment analysis of spatially co-expressed gene clusters on 22 Visium datasets. a** The comparison of enrichment significance over 100 gene clusters detected among all genes from the raw data and the imputed data generated by 3 AE-based models SEDR, STAGATE, and GraphST as well as four tensor-based models CoSTCo, DTD, FIST, and GNTD at their best ranks in each of the 22 Visium datasets. **b** The comparison of enrichment significance over the detected 100 gene clusters at different ranks in the 22 Visium datasets. Source data for (**a**) and (**b**) are provided as a Source Data file.

In the imputation evaluation, GNTD constantly shows the best spot-wise and gene-wise imputation performance with the lowest MAE and MAPE and the highest $R^2$ compared to all other baseline models (Fig. 8a). Similarly, we also show the evaluation of imputation robustness in the 10-fold cross-validation on these 3 Stereo-seq datasets in Supplementary Fig. S4. The analysis of hyper-parameters for GNTD in both spot-wise and gene-wise imputation evaluation on the Stereo-seq data is highly consistent with the results on the Visium data (Fig. 8b). This consistency again supports that a relatively higher rank and properly weighted prior knowledge ($\lambda = 0.1$) of spatial relations among spots and functional relations among genes improve the imputation performance of GNTD. We next applied GNTD to identify spatial domains on the mouse brain coronal hemibrain section and mouse olfactory bulb section, GNTD (ARI = 0.55 and 0.32) clearly outperforms all other models in spot clustering and detects continuous spatial domains that match annotated tissue regions (Fig. 8b). Furthermore, GNTD is able to accurately outline complicated anatomical regions while the baseline models tend to produce over-smoothed spatial domains that obfuscate fine-grained structures (Fig. 8c, d). For example, GNTD isolates the cornu ammonis area 3 (CA3) region from the dentate gyrus (DG) and molecular layer of dentate gyrus (MLDG) regions while the baseline models over-smooth them as one region in the mouse brain coronal hemibrain section. GNTD also demarcates the granule cell layer (GCL-I) and GCL-D regions while the baseline models merge them as one region in the mouse olfactory bulb section. We also visualize the raw data and the imputed data by two-dimensional UMAP in the bottom row of Fig. 8c, d. It is clearly visible that spots in the same tissue region are projected tightly together with good separation from the spots in other tissue regions in the UMAP on GNTD imputed data. Notably, GNTD could depict the spatial trajectory for the mouse olfactory bulb section in the UMAP space, which is consistent with the developmental sequence within the laminar organization starting from

the external plexiform layer (EPL), proceeding bilaterally outwards to the mitral cell layer (MCL) and glomerular layer (GL), olfactory nerve layer (ONL), and then developing the granule cell layer (GCL) lastly. Overall, all these results confirm the strength of GNTD in imputation for high-resolution spatial transcriptomics data.

## GNTD imputation reveals true gene spatial patterns in both low- and high-resolution spatial transcriptomics data

We visualized the expression profiles of 12 known layer-specific marker genes in the Visium raw data and the imputed data generated by the tensor-based models for the DLPFC 151673 section. GNTD imputation properly enhances the expression and enriches the correct cortical laminae validated by ISH data from the Allen Human Brain Atlas (Fig. 9a). While the imputation by CoSTCo and DTD also strengthens expression signals in most of the genes, the imputation remains noisy and lacks spatial continuity, and even obscures original spatial patterns in the raw data. FIST was also unable to preserve the original spatial patterns in the raw data, which suggests that multilinear modeling alone is insufficient to model the complex interactions in the spatial transcriptomics data. We also visualize the expression profiles of 12 known region-specific marker genes in the raw Stereo-seq data and the imputed data generated by the tensor-based models for the mouse olfactory bulb section (Fig. 9b). Similarly, GNTD imputation correctly amplifies the expression signals of the marker genes in their anatomical regions validated by ISH data from the Allen Mouse Brain Atlas. While the imputation by CoSTCo and DTD shows certain correspondence to the anatomical regions for most of the genes as well, the imputation is often so fragmented and over-spread that the original spatial patterns in the raw data are lost. FIST was incapable of improving the original spatial patterns in the raw data, which confirms the importance of nonlinear modeling for spatial transcriptomics data imputation. Collectively, these results illustrate that GNTD is capable

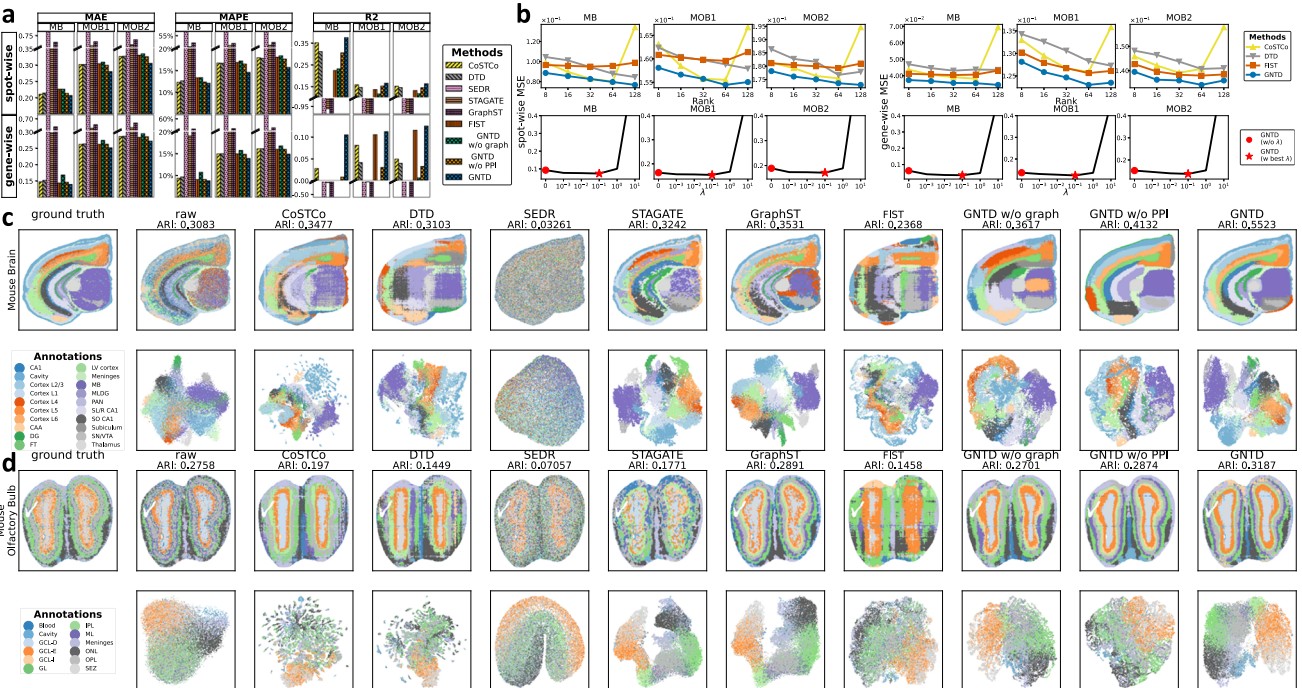

**Fig. 8 | Experiments on 3 stereo-seq spatial transcriptomics datasets.**
**a** Evaluation of imputation accuracy by MAE, MAPE and R². The 3 AE-based deep learning models SEDR, STAGATE, and GraphST, as well as 4 tensor-based models CoSTCo, DTD, FIST, and GNTD, are compared. Each bar shows the mean of the imputation performance over all the spots or all the genes. The results are reported for spot-wise 10-fold cross-validation in the top plot and gene-wise 10-fold cross-validation in the bottom plot. **b** Analysis of hyper-parameter tuning by spot-wise and gene-wise 10-fold cross-validation. Comparison of MSE by varying the rank in tensor decomposition is shown in the plots in the top row, in which each dot shows the mean MSE over all the spots in spot-wise evaluation or all genes in gene-wise evaluation. MSE of GNTD at the best rank by varying $\lambda$ is shown in the plots in the bottom row, in which each dot shows the mean MSE over all the spots in spot-wise evaluation or all genes in gene-wise evaluation. **c** Ground-truth of tissue regions and the detected spatial domains by clustering (Upper panel) and the UMAP embeddings of the spots by highly variable genes (Lower panel) of the raw data and the imputed data by the compared methods at the best rank on the mouse brain section. **d** Ground-truth of tissue regions and the detected spatial domains by clustering (Upper panel) and the UMAP embeddings of the spots by highly variable genes (Lower panel) of the raw data and the imputed data by the compared methods at the best rank on the mouse olfactory bulb section. Source data for (**a**) and (**b**) are provided as a Source Data file.

of revealing the complete gene spatial patterns by the imputation of the raw spatial transcriptomics data. Note that all these tensor models do not alter the scale of the expression by the nature of the MSE-type loss functions. We rescaled the color range based on the minimal and maximal expression for raw and imputed data in Fig. 9 to highlight the spatial patterns with better contrast. The marker gene visualization in the same color range of the original values for both the raw data and the imputed data are also shown in Supplementary Fig. S8.

## Discussions

The focus of the learning task in this research work is on imputing the missing/incomplete gene expressions that fell through the capture for completing the transcriptome-wide gene expressions in the measured tissue locations. This formulation is different from other imputation or imputation-related tasks, which are often augmented with tissue staining images or scRNAseq data beyond spatial gene expression data[16,17,33].

There are methods that aim to match sparse spatial transcriptomics data with scRNAseq profiles[15,34], where spatial transcriptomics data can be reconstructed from the scRNAseq data by deconvolution. The additional assumption for deconvolution is that well-matched scRNAseq data on the same cell population also exist. This assumption might introduce other uncertainties and hinder the interpretation of the downstream analysis to be less relevant to the specific spatial gene expression data. In an additional experiment (see details in the supplementary document), we also compared GNTD with one spatial transcriptomics data deconvolution method Tangram[15] with the same 10-fold cross-validation evaluation of

imputing gene expressions in a coronal region cropped from the same mouse brain Visium dataset used in this study. The result shows that the spatial gene expressions imputed based on the aggregation of scRNAseq data have a very different nature and exhibit low agreement with the original spatial transcriptomics data in both the raw expression values and the correlations, which suggests that this external imputation might not be generally applicable to recovering missing or incomplete gene expressions in the spatial transcriptomics data.

There are also methods proposed for imputing gene expressions in the locations that are not covered by the arranged capture spots[17] or directly improving the resolution of the spatial transcriptomics data by integrative analysis with H&E image data[16,35]. While GNTD does not target these aims directly, a potential follow-up study of this work is to design an extension or a post-processing step to infer additional spatial gene expressions based on spatial proximity. We also investigated running XFuse[16] and ST-Net[35] for full imputation of spatial transcriptomics data. While these methods have been shown to perform well in imputing a small number of genes in the original studies, the applicability to the whole transcriptome seems to be rather limited in both the scalability and lack of a complete evaluation. A more sophisticated experimental design is necessary for a thorough comparison.

Furthermore, even if the AE-based methods[23–25], focusing on extracting the embedding during reconstructing spatial gene expression, can also be adapted to whole transcriptome imputation, the performance is often sub-optimal since the models are optimized to learn low-dimensional embedding smoothing over the spatial

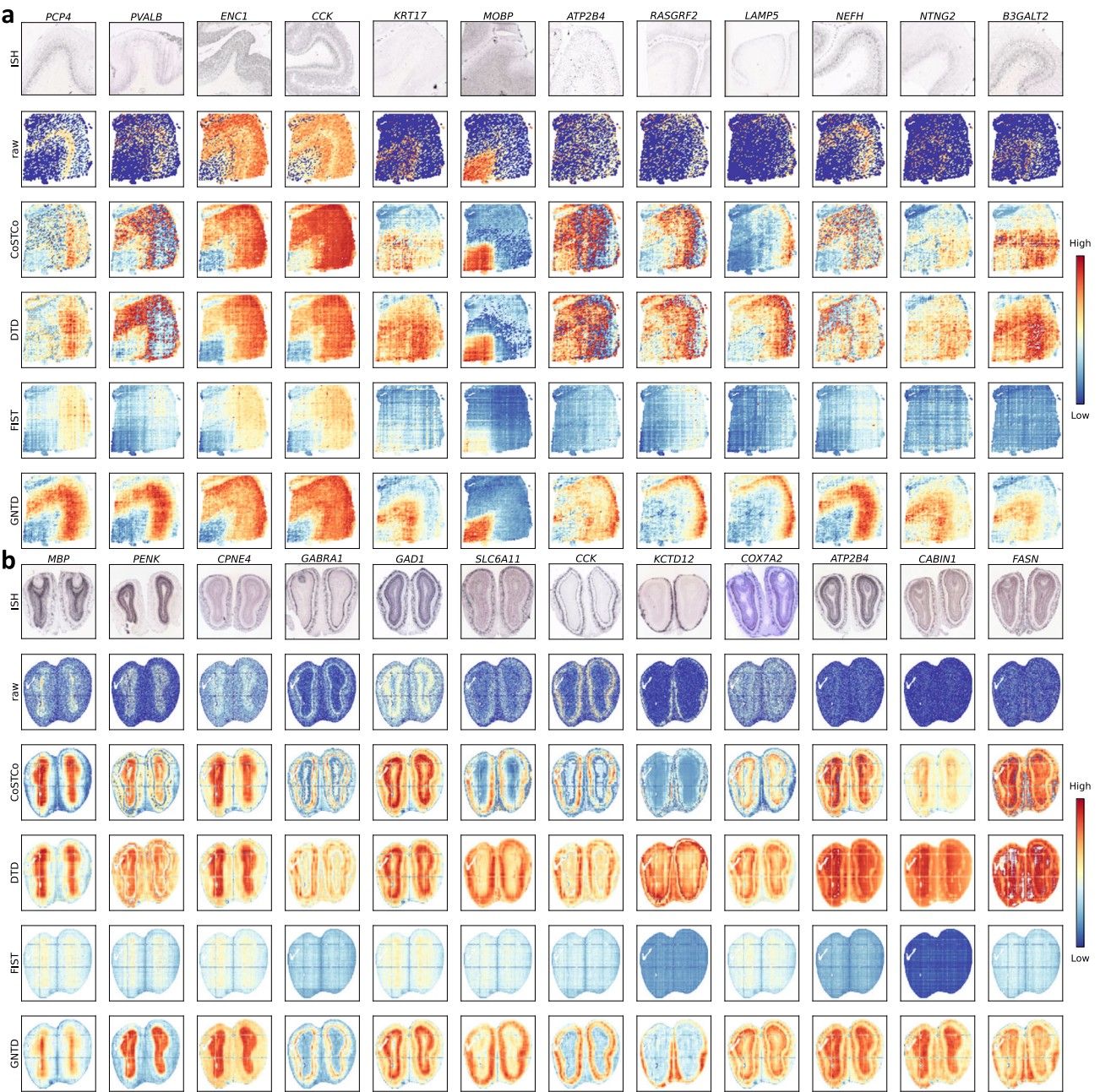

**Fig. 9 | Imputation for recovering the spatial patterns of marker genes on both Visium and Stereo-seq data. a** The visualizations of 12 layer-specific marker genes (*PCP4*, *PVALB*, *ENC1*, *CCK*, *KRT17*, *MOBP*, *ATP2B4*, *RASGRF2*, *LAMP5*, *NEFH*, *NTNG2*, and *B3GALT*) of ISH images, raw data, and imputed data generated by four tensor-based models CoSTCo, DTD, FIST, and GNTD at their best rank on the DLPFC 151673 section from Visium. **b** The visualizations of 12 region-specific marker genes (*MBP*, *PENK*, *CPNE4*, *GABRA1*, *GAD1*, *SLC6A11*, *CCK*, *KCTD12*, *COX7A2*, *ATP2B4*, *CABIN1*, and *FASN*) of ISH images, raw data, and imputed data generated by four tensor-based models CoSTCo, DTD, FIST, and GNTD at their best rank on the mouse olfactory bulb section from Stereo-seq.

neighborhood for some other downstream analyses and it does not necessarily result in better reconstruction on original spatial transcriptomics data. This is supported by the comparison with the three AE-based methods in our experiments.

There have also been many developments of imputation methods for probe-based technologies such as MEMFISH and more recent Nanotring CosMx. The imputation task in this context is to estimate the expressions of the unprofiled genes based on the probed genes and often accompanied scRNAseq data. While GNTD can employ the PPI network to model gene-gene relations to facilitate such gene imputation task, a dense subnetwork is often required for accurate imputation. Apparently, the success will highly depend on which

profiled genes are available for training and which unprofiled genes are targets.

Importantly, our work suggests that imputing spatial transcriptomics data by introducing spatial and functional information in the data itself prior to any analysis consistently improves the standard downstream analyses. Thus, the imputation approach is a more convenient alternative without using other advanced methods for each downstream analysis separately. The denoising nature of the imputation also provides more reliable information for better spatially variable gene detection and potentially better spatially co-expressed gene cluster identification. In addition, there is always a need for analyzing the full spectrum of transcriptome beyond only known marker genes.

For example, transcription-related functions are often performed by lowly expressed genes that can vastly benefit from the imputation[12,36].

Finally, in this study, GNTD has been tested on 10x Visium and Stereo-seq data. While it is possible to apply GNTD to other spatial transcriptomics platforms, such as Slide-seq, MERFISH, and Nano-String CosMx, there are still two limitations. First, the formulation requires the capturing spots to be naturally arranged into a grid-like structure such that the expression profiles can be represented as a tensor. Second, cell segmentation is required to achieve real single-cell and sub-cellular resolution. In the future, we will work on extending GNTD to these transcriptomics platforms by cell binning[37,38] or meshing[39]. In principle, GNTD can also possibly incorporate reference scRNAseq profiles or image-based components from scRNAseq data as default or fixed gene or spatial components in the nonlinear layer. In our future work, we will also probe these possible extensions to further improve the functionality of GNTD.

GNTD is a neural network model for nonlinear tensor decomposition. Its architecture adopts a hierarchical representation by latent features at different levels to capture more complex underlying organization of tensor data, by high-order regularizations with a Cartesian product graph to impose structural relations for avoiding overfitting. These distinct properties of GNTD have been shown to be critical for modeling spatial transcriptomics data for imputation and several other downstream analyses. The results from the extensive experiments over simulations, 22 Visium spatial transcriptomics datasets, and 3 high-resolution Stereo-seq datasets suggest that GNTD is the best method for the imputation of spatially resolved gene expressions by our comprehensive benchmarking and comparison with other methods. The high consistency of the results across all the datasets and between the data from the two spatial profiling platforms also suggests that our findings are highly generalizable to other datasets and potentially, data from other different platforms. The results also demonstrated that the Cartesian product graph constructed from spatial relations among the capturing spots and the functional relations among the genes in the PPI network plays a key role in the imputation performance. Overall, we conclude that GNTD is a useful method for analyzing spatially resolved gene expressions based on a nonlinear tensor completion and high-order graph-regularization by spatial and functional information.

## Methods

### Data preparation and preprocessing

In this study, the experiments focus on spatial gene expression datasets generated with in situ capturing-based spatial transcriptomics technologies, including 22 Visium datasets and 3 Stereo-seq datasets (See details in Supplementary Table S1). The 10x Visium datasets were obtained from two sources. One source contains 10 different mouse and human tissues from 10x Genomics spatial gene expression demonstration[7], among which one human breast cancer tissue was manually labeled with 4 major tumor types and 20 tumor subtypes based on its pathological features by Fu et al.[23]. The other source contains 12 human dorsolateral prefrontal cortex (DLPFC) sections from spatialLIBD project[40], where Maynard et al. have manually annotated all 12 DLPFC sections with up to six cortical layers and white matter based on their morphological features and known spatially variable gene markers. To further demonstrate the applicability to high-resolution spatial transcriptomics data, we also extended the analysis to 3 Stereo-seq datasets of one mouse brain tissue and two mouse olfactory bulb tissues. Chen et al.[10] annotated all 3 tissues with the anatomical regions based on unsupervised spatial clustering and known spatially variable marker genes. For all the datasets, raw unique molecular identifier (UMI) counts were first preprocessed by performing counts per million (CPM) normalization and then log-transformed after adding offset 1.

### Spatial graph and gene graph construction

A spatial graph and a gene graph are constructed to incorporate the prior knowledge of spatial localization of spots and functional relations among genes to guide the spatial transcriptome imputation. The spatial graph models *spatial dependency*—spots in the same spatial neighborhood are more likely to have similar expression profiles; and the gene graph models *functional coherence*—genes within the same functional module such as protein complex are more likely to co-express. We model spatial relations among spots and functional relations among genes by undirected graphs $\mathbf{G}_{xy}$ and $\mathbf{G}_g$. Let $\mathbf{W}_{xy} \in \{0,1\}^{n_x n_y \times n_x n_y}$ be adjacency matrix for $\mathbf{G}_{xy}$, where $[\mathbf{W}_{xy}]_{ij} = 1$ if $i$-th and $j$-th spots are spatially adjacent or similarly expressed otherwise $[\mathbf{W}_{xy}]_{ij} = 0$, and $n_x$ and $n_y$ denote the number of spots along x- and y-axis, respectively. The neighborhood for each spot in $\mathbf{G}_{xy}$ is determined by its 6 nearest spots based on the spot arrangement in the Visium array and its 10 most similar spots based on the gene expression profiles computed by the top 15 PCs of the expression profile. Note that including spot co-expression in spatial graph construction could ameliorate heterogeneity issues within the local neighborhoods for imputation with relatively low-resolution data. Let $\mathbf{W}_g \in \{0,1\}^{n_g \times n_g}$ be the adjacency matrix of $\mathbf{G}_g$, where $[\mathbf{W}_g]_{ij} = 1$ if $i$-th and $j$-th genes are functionally proximate otherwise $[\mathbf{W}_g]_{ij} = 0$, and $n_g$ denotes the number of genes. The functional neighborhood of each gene in $\mathbf{G}_g$ is defined by its connections to other genes based on the protein interactions in the PPI networks. We downloaded the PPI network both for *homo sapiens* and *mus musculus* species from Bio-Grid 4.4, which compiled 1,233,327 and 97,994 interactions respectively. These are mostly experimentally determined physical interactions with high confidence for constituting reliable connections in the PPI networks.

### GNTD

GNTD is a graph-guided nonlinear tensor decomposition model with its architecture outlined in Fig. 1. For any spatial transcriptomics data organized into a 3-way tensor $\mathcal{T} \in \mathbb{R}_+^{n_g \times n_y \times n_x}$, the input of GNTD are 3 index vectors $\mathbf{i}_g \in \mathbb{Z}_+^{n_g}$, $\mathbf{i}_y \in \mathbb{Z}_+^{n_y}$ and $\mathbf{i}_x \in \mathbb{Z}_+^{n_x}$ along gene, $y$ coordinate and $x$ coordinate modes, where a tuple of indexes $(i, j, k)$ of index vectors $\mathbf{i}_g$, $\mathbf{i}_y$ and $\mathbf{i}_x$ can uniquely index an entry $\mathcal{T}_{ijk}$ in the spatial transcriptomics data $\mathcal{T}$, and the output of GNTD is the imputed spatial transcriptomics data $\hat{\mathcal{T}}$. The main components of GNTD are neural tensor decomposition and Cartesian product graph Laplacian regularization. Neural tensor decomposition generalizes tensor decomposition with a neural network to capture the complex nonlinear structures underlying spatial transcriptomics data to impute missing expression values. Meanwhile, Cartesian product graph Laplacian regularization leverages the prior knowledge from both protein-protein interaction graph $\mathbf{G}_g$ and spatial neighbor graph $\mathbf{G}_{xy}$ to guide the expression imputation.

**Neural tensor decomposition.** To better motivate GNTD, we first introduce a general framework for hierarchical tensor decomposition[41–43] and then define the formulation of GNTD under the framework.

**Hierarchical tensor decomposition.** For a rank-$n_r$ CPD decomposition of the 3-way tensor $\mathcal{T} = [\![\mathbf{A}_g, \mathbf{A}_y, \mathbf{A}_x]\!]$, a useful generalization is to impose a $K$-hierarchical structure in each component matrix $\mathbf{A}_m \in \mathbb{R}^{n_m \times n_r}$ as

$$\mathbf{A}_m = \mathbf{W}_m^{(1)}\mathbf{W}_m^{(2)}\ldots\mathbf{W}_m^{(K)}, \forall m = g, y, x, \qquad (1)$$

where $K$ is the number of layers in the hierarchical structure and $\mathbf{W}_m^{(k)}$, $k = 1, 2, \ldots, K$, are dimension-matched sub-factorization matrices of size $(n_m, n_1), (n_1, n_2), \ldots, (n_i, n_{i+1}), \ldots, (n_{K-1}, n_r)$, respectively. Thus, a

2-hierarchical rank-$n_r$ CPD of $\mathcal{T}$ can be defined as

$$\hat{\mathcal{T}} = [\![\mathbf{W}_g^{(1)}\mathbf{W}_g^{(2)}, \mathbf{W}_y^{(1)}\mathbf{W}_y^{(2)}, \mathbf{W}_x^{(1)}\mathbf{W}_x^{(2)}]\!]. \tag{2}$$

To model nonlinearity, nonlinear mappings can be introduced over sub-factorization matrices as,

$$\hat{\mathcal{T}} = f([\![f_g(\mathbf{W}_g^{(1)})\mathbf{W}_g^{(2)}, f_y(\mathbf{W}_y^{(1)})\mathbf{W}_y^{(2)}, f_x(\mathbf{W}_x^{(1)})\mathbf{W}_x^{(2)}]\!]), \tag{3}$$

where $f, f_g, f_y, f_x$ are mapping layers to be defined in the formulation of a neural network. Below, we define the Neural tensor decomposition with the input layer as the *embedding layer*, $f_g, f_y, f_x$ as *nonlinear mapping layers* and $f$ as the *nonlinear aggregation layer*. Note that clearly, hierarchical tensor decomposition can be generalized to any different number of layers in each mode. Practically, only a few layers are needed to capture the representations at each layer depending on the complexity of the data and the amount of available training information. Here, we focus on the simplest 2-layer hierarchy since networks with more layers are much harder to train and lead to no improvement for imputing spatial transcriptomics data.

**Embedding layer.** The embedding layer takes index vector $\mathbf{i}_g$, $\mathbf{i}_y$ and $\mathbf{i}_x$ along gene, $y$ and $x$ modes of the spatial transcriptomics data $\mathcal{T}$ as inputs, and represents these index vector along different modes as latent factor matrices $\mathbf{A}_g \in \mathbb{R}^{n_g \times n_r}$, $\mathbf{A}_y \in \mathbb{R}^{n_y \times n_r}$ and $\mathbf{A}_x \in \mathbb{R}^{n_x \times n_r}$ respectively, where the rank $r$ is shared across all factor matrices. The embedding mapping layer $f^{(\text{emb})}$ can be written as:

$$\mathbf{A}_m = f_m^{(\text{emb})}\left(\mathbf{i}_m; \mathbf{W}_m^{(\text{emb})}\right) = \mathbf{E}_m\mathbf{W}_m^{(\text{emb})}, \forall m = g, y, x \tag{4}$$

where $\mathbf{E}_m \in \{0,1\}^{n_m \times n_m}$ is one-hot embedding matrix of index vector $\mathbf{i}_m$ for mode $m$, $\forall m = g, y, x$. $\mathbf{E}_m = \mathbf{I}_m$. $\mathbf{W}_m^{(\text{emb})} \in \mathbb{R}^{n_m \times n_r}$ are learnable parameters in the embedding layer for each mode $m$, $\forall m = g, y, x$.

The classic tensor decomposition models, such as CPD, could be easily translated into a shallow neural network with 1 embedding layer and reconstruct $\hat{\mathcal{T}}$ based on factor matrices $\mathbf{A}_g$, $\mathbf{A}_y$ and $\mathbf{A}_x$ through multilinear multiplication. However, these multilinear decomposition models cannot handle the needed nonlinearity in the spatial transcriptomics data for modeling arbitrary spatial shapes and gene interactions. Therefore, we next forward the embeddings to the nonlinear mapping layer to learn the nonlinearity within the latent factor matrix for each mode.

**Nonlinear mapping layer.** The nonlinear mapping layer is basically a set of fully connected layers with $n_r$ hidden units for each mode, and takes the factor matrices $\mathbf{A}_g$, $\mathbf{A}_y$ and $\mathbf{A}_x$ from the previous embedding layer as inputs, and outputs the nonlinear latent factor matrices $\tilde{\mathbf{A}}_g \in \mathbb{R}^{n_g \times n_{\tilde{r}}}$, $\tilde{\mathbf{A}}_y \in \mathbb{R}^{n_y \times n_{\tilde{r}}}$, $\tilde{\mathbf{A}}_x \in \mathbb{R}^{n_x \times n_{\tilde{r}}}$, where the rank $\tilde{r}$ is shared across all nonlinear factor matrices. The nonlinear mapping layer applies nonlinear activation function parametric ReLU $\sigma_p(\cdot)$ for each mode. The nonlinear mapping layer $f^{(\text{nlin})}$ can be formally defined as:

$$\tilde{\mathbf{A}}_m = f_m^{(\text{nlin})}\left(\mathbf{A}_m; \mathbf{W}_m^{(\text{nlin})}\right) = \sigma_p(\mathbf{A}_m\mathbf{W}_m^{(\text{nlin})}; a_m), \forall m = g, y, x \tag{5}$$

where $\sigma_p(\cdot) = \max(\cdot, 0) + a_m \min(\cdot, 0)$, $\sigma_p(\cdot)$ is parameterized by $a_m \in [0,1]$ while $\mathbf{W}_m^{(\text{nlin})} \in \mathbb{R}^{n_r \times n_{\tilde{r}}}$ are learnable parameters in the nonlinear mapping layer for all the mode $m$, $\forall m = g, y, x$. Note that the nonlinear mapping layer only models the nonlinearity underlying the latent factor matrix within each mode individually. We then introduce the nonlinear aggregation layer to explore the interactions across the latent factor matrices for different modes.

**Nonlinear aggregation layer.** The nonlinear aggregation layer takes the nonlinear factor matrices $\tilde{\mathbf{A}}_g$, $\tilde{\mathbf{A}}_y$, $\tilde{\mathbf{A}}_x$ as inputs, aggregates them through CPD-like multilinear multiplication, then applies nonlinear

activation function ReLU $\sigma(\cdot)$, and lastly outputs imputed spatial transcriptomics $\hat{\mathcal{T}}$. The aggregation layer $f^{(\text{agg})}$ can be expressed as:

$$\hat{\mathcal{T}} = f^{(\text{agg})}\left(\tilde{\mathbf{A}}_g, \tilde{\mathbf{A}}_y, \tilde{\mathbf{A}}_x; \mathbf{w}\right) = \sigma\left(\sum_i^{n_{\tilde{r}}} \mathbf{w}_i \left[\tilde{\mathbf{A}}_g\right]_{:,i} \odot \left[\tilde{\mathbf{A}}_y\right]_{:,i} \odot \left[\tilde{\mathbf{A}}_x\right]_{:,i}\right), \tag{6}$$

where $\mathbf{w} = \mathbf{w}^{(\text{agg})}$ for simplicity of notations. $\sigma(\cdot) = \max(\cdot, 0)$, $\odot$ denotes the vector outer product, $[\mathbf{A}_m]_{:,i}$ denotes the $i$-th column of $\mathbf{A}_m$, $\forall m = g, y, x$. $\mathbf{w} \in \mathbb{R}^{n_{\tilde{r}}}$ is a learnable parameter to weight nonlinear factor matrices in the nonlinear aggregation layer and $\mathbf{w}_i$ denotes the $i$-th element of $\mathbf{w}$.

**Reconstruction loss.** Given the raw spatial transcriptomics data $\mathcal{T} \in \mathbb{R}_+^{n_g \times n_y \times n_x}$ and the imputed spatial transcriptomics $\hat{\mathcal{T}} \in \mathbb{R}_+^{n_g \times n_y \times n_x}$, $\mathcal{M} \in \{0,1\}^{n_g \times n_y \times n_x}$ is the mask tensor indicating observed entries in the $\mathcal{T}$, where $\mathcal{M}_{ijk}$ is set to be 1 if the $i$-th gene at the coordinates $(j,k)$ has expression in $\mathcal{T}$ and 0 otherwise, we can formally define the reconstructed loss $\mathcal{L}_{\text{recon}}$ for the neural tensor decomposition as:

$$\mathcal{L}_{\text{recon}} = \frac{1}{2}\left|\mathcal{M} \circledast \left(\mathcal{T} - \hat{\mathcal{T}}\right)\right|_F^2 = \frac{1}{2}\left|\mathcal{M} \circledast \left(\mathcal{T} - f_{\text{NTD}}(\mathcal{T}; \mathbf{W})\right)\right|_F^2. \tag{7}$$

Note that the mask matrix $\mathcal{M}$ is optional. When all the entries are considered for training, the loss will be calculated over both the zero and non-zero entries.

**Graph regularization loss.** Given undirected graphs $\mathbf{G}_g$ encoding gene functional modules and $\mathbf{G}_{xy}$ defining spots spatial neighborhood, we can use the Cartesian product graph $\mathbf{G}_c$ combining $\mathbf{G}_g$ and $\mathbf{G}_{xy}$ to impose a regularization over the entries in the imputed tensor $\hat{\mathcal{T}}$ such that the $i$-th gene at the coordinate $(x,y)$ and the $i'$-th gene at the coordinate $(x',y')$ are encouraged to co-express if and only if either the $i$-th and $i'$-th genes are adjacent in the $\mathbf{G}_g$ with the same coordinates (i.e. $(x,y) = (x',y')$) or spots at the coordinates $(x,y)$ and $(x',y')$ are adjacent in the $\mathbf{G}_{xy}$ with the same genes (i.e. $i = i'$). Given the adjacency matrix $\mathbf{W}_g$ of $\mathbf{G}_g$, let $\mathbf{D}_g = \text{diag}(d_1, ..., d_{n_g}) \in \mathbb{R}_+^{n_g \times n_g}$ be the degree matrix of $\mathbf{G}_g$ with $d_i = \sum_j [\mathbf{W}_g]_{ij}$, and $\mathbf{L}_g = \mathbf{D}_g - \mathbf{W}_g \in \mathbf{R}^{n_g \times n_g}$ represents the graph Laplacian for $\mathbf{G}_g$. Similarly, given adjacency matrix $\mathbf{W}_{xy}$ of $\mathbf{G}_{xy}$, $\mathbf{D}_{xy} = \text{diag}(d_1, ..., d_{n_x n_y}) \in \mathbb{R}_+^{n_x n_y \times n_x n_y}$ be the degree matrix of $\mathbf{G}_{xy}$ with $d_i = \sum_j [\mathbf{W}_{xy}]_{ij}$, $\mathbf{L}_{xy} = \mathbf{D}_{xy} - \mathbf{W}_{xy} \in \mathbf{R}^{n_x n_y \times n_x n_y}$ represents the graph Laplacian for $\mathbf{G}_{xy}$. The graph Laplacian for Cartesian product graph $\mathbf{G}_c$ can be expressed as $\mathbf{L}_c = \mathbf{L}_{xy} \oplus \mathbf{L}_g$, where $\oplus$ denotes Kronecker sum. We can further formalize the Cartesian product graph Laplacian regularization as:

$$\mathcal{L}_{\text{reg}} = \frac{1}{2}\text{vec}(\hat{\mathcal{T}})^T \mathbf{L}_c \text{vec}(\hat{\mathcal{T}}) = \frac{1}{2}\text{vec}(\hat{\mathcal{T}})^T (\mathbf{L}_{xy} \oplus \mathbf{L}_g)\text{vec}(\hat{\mathcal{T}}), \tag{8}$$

where $\text{vec}(\cdot)$ denotes the function reshaping the tensor into a vector. However, it is not computationally feasible to obtain the Cartesian product graph Laplacian using $\mathbf{L}_c = \mathbf{L}_{xy} \oplus \mathbf{L}_g$. Alternatively, we need to approximate $\hat{\mathcal{T}}$ with $\tilde{\mathcal{T}} = f'_{\text{NTD}}(\mathcal{T}; \mathbf{W}) = [\![\mathbf{w}; \tilde{\mathbf{A}}_g, \tilde{\mathbf{A}}_y, \tilde{\mathbf{A}}_x]\!]$, and then rewrite the Cartesian product graph Laplacian regularization as:

$$\begin{aligned}
\mathcal{L}_{\text{reg}} &= \frac{1}{2}\text{vec}([\![\mathbf{w}; \tilde{\mathbf{A}}_g, \tilde{\mathbf{A}}_y, \tilde{\mathbf{A}}_x]\!])^T \left(\mathbf{L}_{xy} \oplus \mathbf{L}_g\right)\text{vec}([\![\mathbf{w}; \tilde{\mathbf{A}}_g, \tilde{\mathbf{A}}_y, \tilde{\mathbf{A}}_x]\!]) \\
&= \frac{1}{2}\mathbf{1}_{\tilde{r}}^T(\mathbf{w}\mathbf{w}^T \circledast \tilde{\mathbf{A}}_g^T\tilde{\mathbf{A}}_g \circledast \tilde{\mathbf{A}}_x^T\tilde{\mathbf{A}}_x \circledast \tilde{\mathbf{A}}_y^T\tilde{\mathbf{A}}_y)\mathbf{1}_{\tilde{r}} \\
&\quad + \frac{1}{2}\mathbf{1}_{\tilde{r}}^T(\mathbf{w}\mathbf{w}^T \circledast \tilde{\mathbf{A}}_g^T\tilde{\mathbf{A}}_g \circledast ((\tilde{\mathbf{A}}_x \odot \tilde{\mathbf{A}}_y)^T \mathbf{L}_{xy}(\tilde{\mathbf{A}}_x \odot \tilde{\mathbf{A}}_y)))\mathbf{1}_{\tilde{r}} \\
&\quad + \frac{1}{2}\mathbf{1}_{\tilde{r}}^T(\mathbf{w}\mathbf{w}^T \circledast \tilde{\mathbf{A}}_g^T\mathbf{L}_g\tilde{\mathbf{A}}_g \circledast \tilde{\mathbf{A}}_x^T\tilde{\mathbf{A}}_x \circledast \tilde{\mathbf{A}}_y^T\tilde{\mathbf{A}}_y)\mathbf{1}_{\tilde{r}},
\end{aligned} \tag{9}$$

where $\text{vec}([\![\mathbf{w}; \tilde{\mathbf{A}}_g, \tilde{\mathbf{A}}_y, \tilde{\mathbf{A}}_x]\!]) = (\mathbf{w}^T \odot \tilde{\mathbf{A}}_x \odot \tilde{\mathbf{A}}_y \odot \tilde{\mathbf{A}}_g)\mathbf{1}^T$, $\odot$ denotes Khatri-Rao product and $\circledast$ denotes Hadamard product. The detailed derivation of the regularization term and the gradient are shown in

the supplementary document section S1. Note that we have used similar product graphs in our previous studies in refs. [12,44,45] and have observed very positive results of incorporating the spatial and/or gene functional relations in the models.

**Optimization of GNTD.** With the reconstruction and regularization loss defined, the loss of GNTD can be rephrased as:

$$
\begin{aligned}
\mathcal{L} &= \mathcal{L}_{\mathrm{recon}} + \lambda \mathcal{L}_{\mathrm{reg}} \\
&= \frac{1}{2} \left| \mathcal{M} \circledast \left( \mathcal{T} - f_{\mathrm{NTD}}(\mathcal{T}; \mathbf{W}) \right) \right|_F^2 \\
&\quad + \frac{\lambda}{2} \mathrm{vec}(f'_{\mathrm{NTD}}(\mathcal{T}; \mathbf{W}))^T \mathbf{L}_c \mathrm{vec}(f'_{\mathrm{NTD}}(\mathcal{T}; \mathbf{W})),
\end{aligned}
\tag{10}
$$

where $\lambda$ is the hyperparameter to weight the Cartesian product graph Laplacian regularization for adjusting the impact of prior knowledge in spatial neighbor and PPI graph leveraged in the imputation. The loss function $\mathcal{L}$ can be further minimized by the neural network. We used Adam optimizer in PyTorch with an initial learning rate of 0.05 and trained the model with 90% non-zero entries in $\mathcal{T}$, monitored MSE of remaining 10% non-zero entries for early stopping with 50 epoch patience after first 1,000 epochs. The detailed derivations of the gradient descent steps are provided in the supplementary document section S1.

## Imputation evaluation by cross-validation

We performed both spot- and gene-wise 10-fold cross-validation to evaluate the performance of imputing spatial gene expression on all Visium and Stereo-seq datasets. In the spot-wise cross-validation, all the capturing spots were randomly split into 10 folds, and then the non-zero entries in the capturing spots from 9 folds were used for training and validation while the non-zero entries in the capturing spots from the remaining 1 fold were used for testing. In the gene-wise cross-validation, all the non-zeros entries from each expressed gene were randomly split into 10 folds, and then the non-zero entries pooled in 9 folds were used for training and validation while the non-zero entries pooled in the rest 1 fold were used for testing. Here, since the zeros in the spatial transcriptomics data represent both true biological zeros (not expressed) and a large number of dropouts (not captured), this evaluation focuses on the non-zero entries only for a more precise measure of the prediction performance.

## Spot and gene clustering

We applied mclust[46] to identify spatial domains with spot clustering. mclust is a Gaussian mixture model and has been used for clustering spots in spatial transcriptomics data analysis[24,47]. PCA was performed on the raw data or the imputed data with either highly variable genes or all genes before spot clustering. We empirically selected the top 15 PCs for spot clustering on all the Visium and Stereo-seq datasets. In all the imputed datasets, 15 components capture more than 85% of the variances and are in the range of the numbers achieving the best overall clustering performance for all the methods. The selection of 15 PCs is also consistent with the number of PCs used for clustering the raw data in the previous work on the same Visium datasets[24,47]. We attempted to increase the number of PCs to capture more variance but the performance of spot clustering with mclust was notably worse on most datasets.

To cluster all the genes, we used the commonly used k-means ($k = 100$) to discover co-expressed gene clusters. Similarly, we also performed PCA on the raw data or the imputed data and found that the top 50 PCs generally explain more than 80% variance in both the Visium and Stereo-seq datasets, and provide consistent good clustering results in the datasets. We also varied the number of PCs in gene clustering with k-means but found that fewer PCs resulted in many singleton clusters while more PCs did not provide substantial improvement.

## Evaluation metrics

To compare the imputation performance for spatial transcriptome reconstruction in both Visium and Stereo-seq data, we applied four widely used metrics including, root mean square error (RMSE), mean absolute error (MAE), mean absolute percentage error (MAPE), and coefficient of determination $R^2$ in both spot-wise and gene-wise cross-validations. These metrics are defined as follows,

$$
\begin{aligned}
\mathrm{RMSE} &= \sqrt{\frac{1}{n} \sum_{i=1}^{n} (\mathbf{t}_i - \hat{\mathbf{t}}_i)^2} \\
\mathrm{MAE} &= \frac{1}{n} \sum_{i=1}^{n} \left| \mathbf{t}_i - \hat{\mathbf{t}}_i \right| \\
\mathrm{MAPE} &= \frac{1}{n} \sum_{i=1}^{n} \left| \frac{\mathbf{t}_i - \hat{\mathbf{t}}_i}{\mathbf{t}_i} \right| \\
R^2 &= 1 - \frac{\sum_{i=1}^{n} (\mathbf{t}_i - \hat{\mathbf{t}}_i)^2}{\sum_{i=1}^{n} \left( \mathbf{t}_i - \frac{1}{n} \sum_{i=1}^{n} \mathbf{t}_i \right)^2},
\end{aligned}
\tag{11}
$$

where $\mathbf{t} \in \mathbb{R}^n$ denotes the expression of each spot ($n = n_g$) or gene ($n = n_x \times n_y$) in the original raw spatial transcriptomics data $\mathcal{T}$ while $\hat{\mathbf{t}} \in \mathbb{R}^n$ denotes the expression of each spot ($n = n_g$) or gene ($n = n_x \times n_y$) from the imputed spatial transcriptomics data $\tilde{\mathcal{T}}$ after combining the predictions of each fold in the cross-validation.

To evaluate the imputation performance for spot clustering in both Visium and Stereo-seq data, we mainly used the adjusted rand index (ARI) to quantify the spot clustering accuracy between spatial domains $\{\mathbf{D}_1, \ldots, \mathbf{D}_i, \ldots, \mathbf{D}_{n_c}\}$ identified by the imputation and tissue regions $\{\mathbf{R}_1, \ldots, \mathbf{R}_j, \ldots, \mathbf{R}_{n_c}\}$ defined in the ground truth. ARI is defined as follows,

$$
\mathrm{ARI} = \frac{\sum_{ij} \binom{n_{ij}}{2} - \left( \sum_i \binom{a_i}{2} \sum_i \binom{a_i}{2} \right) / \binom{n}{2}}{\frac{1}{2} \left( \sum_i \binom{0.0pta_i}{2} + \sum_i \binom{a_i}{2} \right) - \left( \sum_i \binom{a_i}{2} \sum_i \binom{a_i}{2} \right) / \binom{n}{2}},
\tag{12}
$$

where $n_{ij}$ denotes the number of common spots between spatial domain $D_i$ and tissue region $R_j$, then $a_i = \sum_j n_{ij}$ indicates the total number of common spots between $D_i$ and all $R_j$ while $b_j = \sum_i n_{ij}$ indicates the total number of common spots between $R_j$ and all $D_i$, and $n$ is the total number of spots overlapped with entire tissue.

To measure the imputation performance for gene clustering on both Visium and Stereo-seq data, we computed the log of the $q$-value of the most significant enriched GO term for each gene cluster and then averaged these minimal q-values across all gene clusters to evaluate the overall enrichment significance. We performed enrichment analysis over 10,185 GO terms from the C5 collection in the Molecular Signatures Database (MSigDB, v2023.1), which includes 7751 biological process (BP) terms, 1009 cellular component (CC) terms, and 1772 molecular function terms. We calculated q-values by adjusting enrichment p-values by false discovery control (FDR) with the Benjamini-Hochberg (BH) procedure.

## Compared methods

We compared GNTD with six methods by their performance of imputation and several downstream analyses on the Visium and Stereo-seq datasets. These methods include three tensor-based models FIST[12] (v1.0.0), CoSTCo[20] (v1.0.0) and DTD[21,22] (v0.1.0), and three Autoencoder-based models SEDR[23] (v1.0), STAGATE[24] (v1.0.0) and GraphST[25] (v1.0.0).

FIST[12] is a graph-regularized linear tensor decomposition model designed for spatial transcriptomics data imputation, which explicitly

leverages both spot spatial relations and gene functional relations to regularize non-negative CP decomposition. In all comparisons, we optimized FIST by using the multiplicative updating rule, then we trained the model with all non-zero entries and halted the training when either the factors residues smaller than $1e-4$ or the total number of epochs reaching 500. Note that we only reported the results from FIST with hyper-parameter weight on the Cartesian product graph equal to 0.01 since it generally performed the best in the imputation compared with other values as reported previously[12].

CoSTCo[20] is a nonlinear tensor decomposition method for sparse tensor completion. CoSTCo learns a nonlinear function among tensor factors with a convolutional neural network (CNN), and shares the parameters in the CNN to preserve the low-rank structure for tensor factors to avoid overfitting on sparse tensors. To apply CoSTCo in the comparisons, we used CoSTCo with the same network architecture as the original paper and trained the model with Adam for 50 epochs, where we set the learning rate to 0.01 and batch size to 128. We randomly selected 90% and 10% non-zero entries for training and validation and stopped the training when MSE on validation not reducing after 10 epochs.

DTD[21,22] is a nonlinear tensor decomposition method for general tensor completion based on multilayer perceptron (MLP) to model nonlinear interaction among tensor factors. We implemented DTD with different numbers of MLP layers and trained these models with Adam for 50 epochs, where we fixed the number of hidden units as rank in all MLP layers and set the learning rate to 0.01 and batch size to 128. We split non-zero entries into 90% and 10% for training and validation as well and performed early-stopping over MSE on validation after 10 epochs. Note that we only reported the results from DTD with 2 MLP layers since it generally shows the best performance in imputation compared with other DTD variants with more layers.

SEDR[23] was proposed to mainly extract low-dimensional latent representations of gene expression embedded with spatial information from spatial transcriptomics data. SEDR employs variational autoencoder (VAE) and variational graph autoencoder (VGAE) to reconstruct gene expression and spatial graph jointly, and could potentially impute the missing expression during the reconstruction. In all comparisons, we followed the original study and built the spatial graph by choosing 10 nearest neighbors for each spot based on its spatial coordinates, and used the same network architecture reported in the original study but tuned the number of hidden units in VAE and VGAE by monitoring the MSE on the validation set.

STAGATE[24] learns a low-dimensional latent representation of gene expression encoding spatial information in spatial transcriptomics data via graph attention autoencoder (GAT), where the spatial graph and cell type-aware graph for STAGATE was also constructed using spot neighborhood and co-expression information. In all the comparisons, we adopted the network architecture and optimized the number of hidden units in GAT by examining the MSE on the validation set. In terms of weight on the cell type-aware graph in attention, we followed the STAGATE tutorial to disable the cell type-aware graph for all 12 DLPFC Visium spatial transcriptomics, otherwise, we set the weight on the cell type-aware graph to 0.5.

GraphST[25] employs graph autoencoder (GAE) with contrastive learning to further accentuate low-dimensional latent representation under local spatial context, where the spatial graph was defined by three nearest neighbors graph based on spatial coordinates. In all the comparisons, we preserved the network architecture from the original study and selected the number of hidden units in GAT by minimizing the MSE on the validation set.

It is very important to note that the three AE-based models, STAGATE, SEDR, and GraphST are trained with all the entries in their loss function. We found that this setting generally works well in all the experiments except for the imputation evaluation focusing only on the non-zeros entries. Thus, for a complete comparison, we trained all three models with both all entries or non-zero entries settings and reported better results in all the comparisons.

## Implementation, running environment, and running time

GNTD is implemented in Python 3.8.12, which requires Numpy 1.21.5, Scipy 1.10.1, Pandas 1.2.3, Scikit-learn 1.0.2, Pytorch 1.10.2, Tensorly 0.6.0, Anndata 0.8.0 and Scanpy 1.9.1. The experiments were conducted on a cluster equipped with AMD Milan 7763 64-core processor, 128GB RAM, and NVIDIA A100 Tensor Core GPU. In this environment, GNTD requires ~15 min of wall time on the Visium data with around 5k spots and 20k genes and roughly 40 minutes of wall time on the Stereo-seq data with around 50k spots and 10k genes. GNTD consumes around 4GB and 15GB of GPU memory, respectively when running on the Visium data and the Stereo-seq data.

## Statistics and reproducibility

We performed standard 10-fold cross-validation to evaluate the imputation performance in both spot-wise and gene-wise experiments. All the spots with sufficient gene expressions (minimal 1,438 spots among all the datasets as shown in Supplementary Table S1) are randomly split into 10 folds and the spots with few or no expressions are excluded in the spot-wise experiment. Non-zero entries from all the expressed genes (minimal 9,557 genes among all the datasets as shown in Supplementary Table S1) are split into 10 folds by using a stratified strategy and the genes that are expressed at less than 10 spots are excluded in the gene-wise experiment.

We applied the Python package Scanpy (v1.9.1) to identify differentially expressed genes among spatial regions. The P-values in the differential expression analysis were all calculated using the Wilcoxon rank-sum test. k-means function from the Python package scikit-learn (v1.1.3) was used to detect gene clusters. R package clusterProfiler (v3.12.0) was used to perform the enrichment analysis of the differentially expressed genes and the gene clusters. The P-values in the enrichment analysis were all calculated using the one-sided hypergeometric test. All the spots and expressed genes in the imputation experiments were retained and no spots or genes were excluded for all the downstream analyses.

## Reporting summary

Further information on research design is available in the Nature Portfolio Reporting Summary linked to this article.

## Data availability

The datasets analyzed in this paper are available in raw-form from their original studies. Specifically, ten Visium spatial transcriptomics datasets for five mouse brain tissues, one mouse kidney tissue, two human breast cancer tissues, one human heart tissue, and one human lymph node tissue are collected from the 10x Genomics website https://support.10xgenomics.com/spatial-gene-expression/datasets/, where the manual annotation on the human breast cancer section 1 is accessible at https://github.com/JinmiaoChenLab/SEDR_analyses/tree/master/data/BRCA1. Twelve Visium spatial transcriptomics datasets for human dorsolateral prefrontal cortex and their manual annotations are obtained from the LIBD project http://spatial.libd.org/spatialLIBD/. Three Stereo-seq spatial transcriptomics datasets for one mouse brain and two mouse olfactory bulb tissues and their manual annotations are available at https://db.cngb.org/stomics/mosta/download/. Source data are provided with this paper.

## Code availability

GNTD is implemented in Python and the code is publicly available through GitHub at https://github.com/kuanglab/GNTD. The code can also be accessed through Zenodo at https://doi.org/10.5281/zenodo.10063263[48].

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

## Acknowledgements
This research work is supported by a grant from the National Science Foundations, USA (NSF BIO DBI-IIBR 2042159).

## Author contributions
R.K. conceived and supervised the study. T.S. and R.K. developed the computational method. T.S. implemented the software. R.K. and C.B. generated the simulation data. T.S., C.B. and R.K. conducted experiments and performed analyses on the simulation and real data. T.S. and R.K. interpreted the real data. T.S. and R.K. wrote the manuscript. All authors read and approved the final manuscript.

## Competing interests
The authors declare no competing interests.
