## [Peer Review File · Nature Communications]

GNTD: Reconstructing Spatial Transcriptomes with Graph-guided Neural Tensor Decomposition Informed by Spatial and Functional RelationsReviewer #1 (Remarks to the Author):

In this study, the authors develop GNTD, a computational method to perform imputation for in-situ capturing based spatial transcriptomics. The method uses neural tensor decomposition to capture non-linear relationships among cells/spots. To avoid overfitting, the authors use a graph regularization informed by the spatial relationship between cells and the functional relationship between genes. Through evaluation using simulated and real data, the authors show that their method (GNTD) outperforms other methods in spatial gene expression imputation, spatial domain detection, cell/spot clustering, and spatially co-expressed gene cluster interpretation. While the presentation and writing appear clear there are a few issues that require attention:

1. Evaluation based on simulation (section 2.2):

(1) The authors used 40% and 80% for zero inflation. What is the rationale for using these two values? Are they based on empirical data? For the simulation result to be useful, the zero inflation percentage needs to be reasonably similar to reality. Although ground truth is not available, in-situ hybridization based spatial transcriptomics can be used as a proxy to estimate zero inflation percentage given their high capturing efficiency.

(2) For the evaluation of the performance of spatial gene pattern recovery (line 177), what is the AUC for 50 ubiquitously expressed genes? How many of them are considered spatially variable after imputation using different methods? One major concern for imputation methods in general is that although they can enhance true positives, they may create false positives. The authors need to show that their method can control for false positives.

(3) Similarly, how many true spatial variable genes are detected as spatially variable (true positive) using different methods?

2. Section 2.3

(1) What is the rationale for only using non-zero entries for training, validation, and testing? Some of the zero entries could be true zeros. Therefore, it would be informative to see the same analysis with zero entries.

(2) For spot-wise imputation, what is the proportion of non-zero entries among all entries for each data? Similarly, for gene-wise imputation, what is the average proportion of non-zero entries for each gene over all datasets? The concern here is that 10% non-zero entries may not be enough to get a stable testing result.

3. Section 2.4

(1) For the evaluation result in Fig. 6, in the original STAGATE and SEDR paper, they have performed similar evaluations and both of them were able to recover the six layers (Fig. 2 in the original STAGATE paper and Fig 2&4 in the original SEDR paper). They even have higher ARI than GNTD (0.6 for STAGATE and 0.573 for SEDR compared to 0.543 for GNTD). But they seem to perform poorly here. Why is that? What is the evaluation and implementation difference between this paper with the original papers of STAGATE and SEDR?

(2) For the clustering result in Fig. 6C, the clustering pattern of DTD and even raw data also reflect spatial trajectory that agrees with the chronological order of cortex layer development. The authors should modify their statement from line 302 to 308 to reflect this.

(3) Also, the fact that most methods perform worse than raw data except for GNTD creates questions like: (1) is imputation really necessary on real data, (2) is GNTD over smoothing the data. The graph regularization based on spatial and gene relationships assumes local smoothness, which is not true of many cell types. For example, although the isocortex has this six-layer structure, there are still cell types (for example, inhibitory neurons) that are more sparsely distributed across layers. This information seems to be smoothed out after GNTD imputation. This gives rise to a higher ARI since the six-layer structure is used as ground truth but it will have a negative impact on downstream analysis and new discoveries. It will be better to show the spatial expression pattern of a few marker genes of inhibitory neurons before and after imputation as a sanity check and design some tests to show that the local smoothness assumption won't remove sparse signals. The same thing applies to other results like Fig. 8D where there are sparsely distributed cells in the ground truth data but in the imputed data, they are gone and every spatial domain seems very homogenous.

(4) Regarding the statement about surrounding sub-regions and core regions (line 317), it is better to label them on the plot so we know what is considered as core region and what is

considered as surrounding region. In addition, is the enrichment analysis performed between core and surrounding regions within each homogeneous spatial region defined by previous annotation, or performed on all surrounding regions/core regions pulled together? Table 1 only has one set of GO terms for core regions and surrounding regions, respectively. It would be good to see this enrichment comparison performed separately for every homogenous region that got dichotomized.

4. Section 2.5

More significant enrichment doesn't necessarily mean an improvement. What really matters is that the enriched GO terms actually reflect true biology. Admittedly, it is hard to quantify. But at least showing the top enriched GO terms for each method of each data could be informative. The authors can discuss the enrichment results of a few datasets in the main text and provide the full enrichment result in a supplementary file.

Minor points:

1. To construct a PPI network, are all interactions in one species used, or only a subset of interactions in the target tissue are used? Ideally using interactions in the target tissue would be better if possible.
2. λ and rank first show up in Fig. 2 but they are formally introduced in section 2.3 (line 231). The meaning of λ and rank should be introduced earlier.
3. Fig 4 and 5: when the authors say "This degradation is more significant in the sparser datasets of the 12 DLFC sections.", it will be better to label those sparse sections for better visualization
4. Fig. 9: the author should include ISH data from the Allen Human Brain Atlas in the figure so that it is easier for people to evaluate the imputation.

Reviewer #2 (Remarks to the Author):

Summarized Comments:

The authors proposed a new spatial transcriptome (ST) method that overcomes several limitations in current ST technologies, such as the sparseness of measured expression levels. Their approach utilizes tensor decomposition by modeling spatial dependency (i.e., neighboring spots have similar expression patterns) and functional coherence (i.e., genes with the same functional module like PPI have similar expression patterns). While the idea is good, it is not a novel way to model spatial positional information since modeling dependency was proposed in Science 2019 at the time of the emergence of ST technologies (PMID: 30948552; no mentioning).

Although the authors compared their method with many other approaches, it is unclear whether the graph-based tensor decomposition is the best approach since they did not compare it with convolutional neural network-based approaches. It would be better if they could clearly show the computational cost of their method and compare it with other methods to demonstrate its applicability. Moreover, without GPU, running their method may be difficult.

Overall, I think the manuscript requires additional analyses and a more detailed description to be suitable for publication in this journal.

Major comments:

The authors compared their methods with three other tensor-based models and two autoencoder-based models; however, they did not compare it with CNN-based models, such as those described in PMID 32572199, which incorporate image information from H&E staining as the input dataset. Additionally, XFuse (PMID: 34845373), which uses a deep generative model, is also an important method that could be compared to their proposed approach. Comparing these methods would be essential to determine the novelty of the proposed method.

Their analyses did not take into account the resolution of spatial measurement, as noted in PMID: 35260632. While the figures in the manuscript suggest that predicted values fill the entire slide, this may be a misunderstanding for readers. If I understand correctly, their method cannot fill in unmeasured regions among measured spots. This should be mentioned as a limitation in the main

text.

It was not clear how the authors selected their training, validation, and testing datasets. Then, I'm not sure these are fair comparisons (although the author said "for a fair comparison"). The authors should provide more details or a flowchart to clarify these procedures.

From the authors' codes, they used Pytorch for their analyses; however, there is no information in the main manuscript. At least version information is required.

- Line 440: Although they used publicly available 10x Visium datasets from the manufacturer's website; however identification is very difficult. I recommend showing more detailed information, such as URL, unique name, or DOI.
- Line 454: I suppose, before log transformation, the authors added pseudo-count values. There are several ways to normalize the UMI counts by library size. Please show exactly.
- Line 470: Why top 15 PCs? Empirical selection in line 602 was also not reasonable. Determining from an elbow plot or some rules (i.e., total explained variance > 95%) is better. Based on lines 608-609, I felt that using 15 PCs is few.
- Line 477: Please explain BioGrid. In my understanding, STRING is a major database for PPI.
- Line 595: What were the criteria for the early stopping?
- Line 606: The parameter for k-means clustering was not provided.
- Line 620: MSE and RMSE return almost the same information. Why did the authors use both?

Minor comments:

- Line 645: Need version information for MSigDB.
- Lines 652 – 653: Need version information for these methods.

Reviewer #3 (Remarks to the Author):

This manuscript presents a graph-guided neural tensor decomposition model named GNTD for reconstructing spatial transcriptomes (ST). GNTD first models the spatial coordinates and gene expressions with a 3-way tensor, then learns nonlinear latent factors to reconstruct the tensor via a multiple layer neural network. Comprehensive assessments on simulated and experimental datasets were conducted to show that GNTD outperformed existing methods in imputing ST data. The authors also demonstrated the effects of data imputation on downstream analysis tasks such as clustering.

1. GNTD adopts tensor decomposition for ST data imputation. However, its novelty is not clear as compared to the previously published FIST method (Li et al.). The frameworks of GNTD and FISH seem to be similar. It is thus important to highlight the differences between GNTD and FIST.
2. In addition to SEDR and STAGATE, several more recent deep learning models such as GraphST (Long et al.) and SpaceFlow (Ren et al.) also output reconstructed spatial transcriptomics. Including them in the comparison would be informative to the readers.
3. In Figure 3, the spatial expression patterns of three example genes were shown. It would be better to indicate the names of the three genes in the figure. How were these three genes selected? Or they are simply three random genes.
4. Most probe-based spatial transcriptomics technologies do not cover the whole transcriptome. For example, the cosmx technology by Nanostring is only able to probe 960 genes. Is it possible for GNTD to extend the 960 genes to the whole transcriptome if scRNA-seq data from the matching tissue type are available?
5. Some imputation methods such as Tangram achieve ST reconstruction by integrating a reference scRNA-seq data with ST. I wonder whether reference-based methods like Tangram or reference-free methods like GNTD would perform better.
6. The data derived from spatial transcriptomics especially those with single-cell or subcellular resolution are believed to be sparse. A plot showing the sparsity level of the tested data before and after imputation with different methods would be helpful. More importantly, it has been observed that some tissue regions are more difficult to penetrate, and thus have higher sparsity level. It would be helpful to visualize the spatial distribution of sparsity before and after imputation.
7. Some imputation can distort the gene expression distribution. Pre- and post-imputation

distribution plots are needed to confirm that the imputation does not introduce data distortion.

8. Figure 7 shows the results of enrichment analysis of spatially co-expressed gene clusters. If possible, it would be helpful to add a correlation heatmap plot to show the spatial co-expression patterns of the gene clusters identified before and after imputation.

9. In Figure 9B, the last three genes showed nearly zero expressions before imputation. However, after imputation their expressions became very high and detected all over the tissue. Such substantial changes need to be validated, for example with RNAscope or multiplex IHC/IF to confirm their expressions.

10. An evaluation of memory usage and runtime with respect to data size can guide users to choose the best method for their data analysis task.

11. When testing SEDR, the authors built the spatial graph among spots by combining spot neighbourhood and co-expression information. However, in the original SEDR model, only spot neighbourhood is used to build the graph. Such inconsistency may lead to poorer performance of SEDR.

12. To evaluate the imputation performance, the authors compared five methods, including three tensor-based models CoSTCo, DTD, and FIST, and two deep learning models SEDR, and STAGATE. But only the tensor-based models were compared with GNTD on the simulated data. The authors should justify the reasons.

13. Figures 3 and 8 show negative values for both SEDR and STAGATE in terms of the R2 metric. Further explanations are required to clarify this.

14. In the clustering analysis, some results look a bit weird, such as the results of CoSTCo, STAGATE, and FIST on slice 151673 (Figure 6), and results of STAGATE on mouse brain Stereo-seq data. The authors should double-check their codes and ensure the accuracy of all results presented in the manuscript.

15. In the method section, the symbols and functions need to be clearly defined. For example, symbols such as n_g , n_y , and n_x are undefined. What $f_m()$ and $f_{agg}()$ represent should be described.

16. The authors claim that they trained deep learning methods SEDR and STAGATE with non-zero entries to do the imputation. However, both SEDR and STAGATE typically take the whole gene expressions as input features during model training by default. It should be noted that using non-zero entries for training these models may impact their performance since the default parameters assume whole gene expressions as inputs.

Reference

[1]. Li, et al. Imputation of spatially-resolved transcriptomes by graph-regularized tensor completion. *PLoS computational biology* 17(4), 1008218 (2021).

[2]. Long, et al. Spatially informed clustering, integration, and deconvolution of spatial transcriptomics with GraphST. *Nature Communications* 14.1 (2023): 1155.

[3]. Ren, et al. Identifying multicellular spatiotemporal organization of cells with SpaceFlow. *Nature communications* 13.1 (2022): 4076.

Revision Summary

We sincerely thank the reviewers for the constructive comments on the previous submission for improving our research work. We made substantial revisions in this resubmission in response to the comments. Below is a summary of the major changes that have been made to address the key points in the review, followed by the response to each specific comment from the reviewers.

- Added a more recent deep learning method GraphST (based on generative AE neural network) as a new baseline. The results are reported in imputation cross-validation in Figure 3, spot clustering in Figure 6, gene enrichment analysis in Figure 7, and experiments on stereo-seq data in Figure 8.
- Applied three deep learning methods, GraphST, SEDR, and STAGATE by training with all entries or only non-zero entries in the simulation and the experiments on the real datasets. The methods are also tuned for the optimal number of hidden units. The results are reported in Figure S1, Figure 3, Figure 6, Figure 7, and Figure 8.
- Included the ISH images of the marker gene expressions in Figure 9 for comparison with the imputed gene expressions.
- Performed a more comprehensive analysis of three surrounding and core tumor regions in the breast cancer tissue. The visualization of the regions is shown in Figure S6 and functional enrichment analysis of the differentially expressed genes is given in Table S2.
- Performed a functional analysis of gene clusters detected in the mouse kidney dataset. The mean expression patterns are shown in Figure S7 and the gene enrichment analysis is reported in Table S3.
- Added new paragraphs to Section 3.3 in the Methods to motivate GNTD by a hierarchical tensor decomposition framework.
- Added a new Section 4 Discussions to discuss other formulations of imputation tasks, and in particular compared the imputation based on scRNAseq and ST data deconvolution with Tangram in the discussion.
- Section 1. Introduction was extended to discuss the previous works that have modeled spatial dependency for ST data analysis.
- Added a new Section 3.8 to provide the information on implementation, running environment, and running time.
- We also included several new results in responses to specific comments for additional experiments. These new results are reported in the supplementary documents,
 - Figure S2: Heatmap visualization of the imputed spatial variables in the simulation.
 - Figure S3: Visualization of all the 50 spatial variable genes in the simulation.
 - Figure S4: Analysis of the variance of the 10-fold cross-validation results to assess the robustness of the cross-validation evaluation.
 - Figure S5: Comparison between the distributions of gene expression in pre- and post-imputation data on 22 Visium and 3 Stereo-seq datasets.
 - Figure S8: Visualization of the imputed marker genes using the original gene expression without rescaling for better visualization.
 - Figure S9: Visualization of summarized expression over spatial patterns for marker genes on the human brain DLPFC section.
 - Figure S10: Analysis of the tissue regions with highly sparse expression in the Visium datasets.

We highlighted all the revised text in the manuscript. Below are the responses to each comment.

Reviewer #1 (Remarks to the Author):

1. Evaluation based on simulation (Section 2.2):

(1) The authors used 40% and 80% for zero inflation. What is the rationale for using these two values? Are they based on empirical data? For the simulation result to be useful, the zero inflation percentage needs to be reasonably similar to reality. Although ground truth is not available, in-situ hybridization based spatial transcriptomics can be used as a proxy to estimate zero inflation percentage given their high capturing efficiency.

Response: Thanks for reminding us of the missing explanation. The percentages of zero inflation were indeed chosen to simulate real data. In particular, 40% dropout gives a density of around 50%, which is close to typical ISH spatial transcriptomics data and 80% dropout gives a density of around 13%, which is close to the density of typical sparser Visium data as we show in Table S1. The explanation along with the references are now included in lines 176-179.

(2) For the evaluation of the performance of spatial gene pattern recovery (line 177), what is the AUC for 50 ubiquitously expressed genes? How many of them are considered spatially variable after imputation using different methods? One major concern for imputation methods in general is that although they can enhance true positives, they may create false positives. The authors need to show that their method can control for false positives.

(3) Similarly, how many true spatial variable genes are detected as spatially variable (true positive) using different methods?

Response: Thanks for the excellent suggestion. We now also included a detailed analysis of the spatial variable gene detection after imputation. The results are reported in Figure S2. GNTD is the only method that does not introduce any false positive or false negative spatial variable genes in the simulation.

2. Section 2.3

(1) What is the rationale for only using non-zero entries for training, validation, and testing? Some of the zero entries could be true zeros. Therefore, it would be informative to see the same analysis with zero entries.

Response: Thanks for raising this critical point about the experimental design. Indeed, there are two kinds of training to learn the neural network models, 1) training with only non-zero entries or 2) all the entries (including zero entries), specified by the implementation of the loss function. Note that here all the entries are always used as input and the discrepancy is in which entries are penalized in the loss function. In this revision, we trained the baseline methods, SEDR, STAGATE, and GraphST with both settings and further optimized the number of hidden units in the neural network. The results are reported in Figure S1. For the tensor completion models, DTD and CoSTCo, the loss is always measured on the non-zero entries only, similarly as they were originally designed for scalable sparse data imputation. Also, note that since the zero entries can be either dropouts or true biological zeros, training with all the zeros does not necessarily improve every learning task in our experiments as it is shown in Figure S1.

(2) For spot-wise imputation, what is the proportion of non-zero entries among all entries for each data? Similarly, for gene-wise imputation, what is the average proportion of non-zero entries for each gene over all datasets? The concern here is that 10% non-zero entries may not be enough to get a stable testing result.

Response: Thanks for the question on the robustness of our 10-fold cross-validation evaluation on the spots. Since there are thousands of spots, each fold contains hundreds of spots with at least tens of thousands of non-zeros entries to calculate the performance measure. To show the stability of the evaluation, we plot the mean and the variance of the performance measures across the 10 folds in Figure S4 on all the datasets. The variance is very small with few outlier values, which suggests the test result is reliable.

3. Section 2.4

(1) For the evaluation result in Fig. 6, in the original STAGATE and SEDR paper, they have performed similar evaluations and both of them were able to recover the six layers (Fig. 2 in the original STAGATE paper and Fig 2&4 in the original SEDR paper). They even have higher ARI than GNTD (0.6 for STAGATE and 0.573 for SEDR compared

to 0.543 for GNTD). But they seem to perform poorly here. Why is that? What is the evaluation and implementation difference between this paper with the original papers of STAGATE and SEDR?

Response: Thanks for raising the point for clarification. This is related to the previous response regarding training with all the entities or only non-zero entries. STAGATE and SEDR are embedding methods, which are not trained for perfect reconstruction. We tuned the models and found out that STAGATE and SEDR work better by training with only non-zero entries for the imputation task but worse for spot clustering. In this revision, we report the results of STAGATE and SEDR trained with non-zero entries in the imputation experiment shown in Figure 3 and the results of STAGATE and SEDR trained with all entries in the spot clustering experiment shown in Figure 4. We believe we have reported the best potential of these methods with optimized training design in the experiments.

(2) For the clustering result in Fig. 6C, the clustering pattern of DTD and even raw data also reflect spatial trajectory that agrees with the chronological order of cortex layer development. The authors should modify their statement from line 302 to 308 to reflect this.

Response: Thanks for pointing out the neglected result in our previous submission. We added a statement about the UMAP visualization of DTD, STAGATE, and GraphST in lines 361-363.

(3) Also, the fact that most methods perform worse than raw data except for GNTD creates questions like: (1) is imputation really necessary on real data. (2) is GNTD over smoothing the data. The graph regularization based on spatial and gene relationships assumes local smoothness, which is not true of many cell types. For example, although the isocortex has this six-layer structure, there are still cell types (for example, inhibitory neurons) that are more sparsely distributed across layers. This information seems to be smoothed out after GNTD imputation. This gives rise to a higher ARI since the six-layer structure is used as ground truth but it will have a negative impact on downstream analysis and new discoveries. It will be better to show the spatial expression pattern of a few marker genes of inhibitory neurons before and after imputation as a sanity check and design some tests to show that the local smoothness assumption won't remove sparse signals. The same thing applies to other results like Fig. 8D where there are sparsely distributed cells in the ground truth data but in the imputed data, they are gone and every spatial domain seems very homogenous.

Response: Thanks for raising the critical question(s). While Indeed, there is a risk of over-smoothing with the graph regularization, the risk can be controlled by tuning the weight parameter of the graph regularization as shown in Figure 5 by cross-validation on the imputation values. Evaluations by detecting spatial variables, clustering, and most importantly RMSE/R² indicate there is no systematic over-smoothing in the imputation with the optimal weight parameter. In addition, the marker gene visualization in Figure 9 also suggests that the smoothing is playing a positive role without any over-smoothing in these marker genes, compared to the ISH images added in this revision (Thanks to the suggestion by this reviewer). For the clustering results in Figure 8D, we believe it is generally different to detect small clusters in scRNAseq or ST RNAseq data, and none of the methods was able to capture the small clusters. Thus, we believe the no detection is not due to over-smoothing.

We also added a discussion for the value of imputation at the end of Section 4. Discussions to emphasize the advantage of imputation-based analysis.

(4) Regarding the statement about surrounding sub-regions and core regions (line 317), it is better to label them on the plot so we know what is considered as core region and what is considered as surrounding region. In addition, is the enrichment analysis performed between core and surrounding regions within each homogeneous spatial region defined by previous annotation, or performed on all surrounding regions/core regions pulled together? Table 1 only has one set of GO terms for core regions and surrounding regions, respectively. It would be good to see this enrichment comparison performed separately for every homogenous region that got dichotomized.

Response: Thanks for pointing out the insufficient analysis of the clustering results on the breast cancer tissue. In this revision, we performed a more comprehensive analysis of three surrounding and core tumor regions. The visualization of the regions is shown in Figure S6 and functional enrichment analysis of the differentially expressed genes is given in Table S2. The results are also discussed in the main text.

4. Section 2.5

More significant enrichment doesn't necessarily mean an improvement. What really matters is that the enriched GO terms actually reflect true biology. Admittedly, it is hard to quantify. But at least showing the top enriched GO terms for each method of each data could be informative. The authors can discuss the enrichment results of a few datasets in the main text and provide the full enrichment result in a supplementary file.

Response: Thanks for the suggestion. Following this suggestion, we performed a functional analysis of gene clusters detected in the mouse kidney dataset. The mean expression patterns are shown in Figure S7 and the gene enrichment analysis is reported in Table S3.

Minor points:

1. To construct a PPI network, are all interactions in one species used, or only a subset of interactions in the target tissue are used? Ideally using interactions in the target tissue would be better if possible.

Response: Since tissue-specific interactions are not available in any species, we are using the PPI network as a background interaction map for potential candidate interactions. While some of the interactions might not be utilized/activated in specific tissues, the functional relations are still useful background knowledge, e.g. functional-related genes are more likely to co-express than random genes.

Potentially, we can replace the PPI network with the co-expression network for tissue-specific co-expression. Unfortunately, our preliminary results did not show any improvement in the performance with reduced efficiency and less robustness. Thus, we did not pursue this alternative further.

2. λ and rank first show up in Fig. 2 but they are formally introduced in section 2.3 (line 231). The meaning of λ and rank should be introduced earlier.

Response: Thanks for pointing out the issue. λ and rank are now mentioned in lines 185-187.

3. Fig 4 and 5: when the authors say "This degradation is more significant in the sparser datasets of the 12 DLFP sections.", it will be better to label those sparse sections for better visualization

Response: Thanks for the suggestion. We now added Table S1 to list the statistics of the datasets and a column named 'Sparsity' specifies the sparsity of the dataset we used in this study.

4. Fig. 9: the author should include ISH data from the Allen Human Brain Atlas in the figure so that it is easier for people to evaluate the imputation.

Response: Thanks for the excellent suggestion. This revision now includes the ISH images of the marker gene expressions in Figure 9 for comparison with the imputed gene expressions in both the Visium and Stereo-seq data. The imputed expressions do match the ISH images with a high agreement.

Reviewer #2 (Remarks to the Author):

Summarized Comments:

The authors proposed a new spatial transcriptome (ST) method that overcomes several limitations in current ST technologies, such as the sparseness of measured expression levels. Their approach utilizes tensor decomposition by modeling spatial dependency (i.e., neighboring spots have similar expression patterns) and functional coherence (i.e., genes with the same functional module like PPI have similar expression patterns). While the idea is good, it is not a novel way to model spatial positional information since modeling dependency was proposed in Science 2019 at the time of the emergence of ST technologies (PMID: 30948552; no mentioning).

Although the authors compared their method with many other approaches, it is unclear whether the graph-based tensor decomposition is the best approach since they did not compare it with convolutional neural network-based approaches. It would be better if they could clearly show the computational cost of their method and compare it with other methods to demonstrate its applicability. Moreover, without GPU, running their method may be difficult. Overall, I think the manuscript requires additional analyses and a more detailed description to be suitable for publication in this journal.

Response: Thanks for the helpful suggestions and comments. Thanks for bringing our attention to the previous work in Science, which introduced conditional autoregressive prior (equivalent to Laplacian graph-based likelihood) for a generalized linear model with zero-inflated Poisson link function. We included a background review for using spatial dependence for ST data analysis in Section 1. Introduction, to better explain the context of this research work. As suggested by the reviewers, we added a more recent deep learning method GraphST (based on a generative AE neural network) as a new baseline. We also included a new Section 3.8 to provide information on implementation, running environment, and running time, and discuss the applicability and limitations of GNTD in Section 4. Discussions.

Major comments:

The authors compared their methods with three other tensor-based models and two autoencoder-based models; however, they did not compare it with CNN-based models, such as those described in PMID 32572199, which incorporate image information from H&E staining as the input dataset. Additionally, XFuse (PMID: 34845373), which uses a deep generative model, is also an important method that could be compared to their proposed approach. Comparing these methods would be essential to determine the novelty of the proposed method.

Response: Thanks for the suggestions. We added a more recent deep learning method GraphST (based on a generative AE neural network) as a new baseline. The results are reported in imputation cross-validation in Figure 3, spot clustering in Figure 6, gene enrichment analysis in Figure 7, and experiments on stereo-seq data in Figure 8.

Regarding the comparison between GNTD and the H&E image-based deep learning models that could potentially perform the imputation task, such as CNN-based model ST-Net (PMID 32572199) and generative model XFuse (PMID: 34845373), while we completely agree that it is promising to explore additional information (such as nuclei segmentations, convolutional features) in H&E image for imputation, after working with the two packages, we feel it is difficult to design a meaningful comparison with either ST-Net or XFuse as part of this study due to the following reasons:

For ST-Net,

1. ST-Net only relies on convolutional features extracted from DenseNet to predict the expression of the top 250 genes (highly variable genes) rather than the whole transcriptome. It is actually unclear if ST-Net is sufficiently expressive to predict the expression of more than 10k genes with the 1,024 convolutional features. This is a fundamentally different setting compared with GNTD which models non-zero entries in the spatial gene expressions for the whole transcriptome-wide imputation.
2. It is also not reasonable to apply GNTD to impute fewer genes like ST-Net because the GNTD performance might be undermined by involving just such a small subset of genes.

3. ST-Net also needs to be trained with as many H&E image patches over spots as possible, and it used around 20k patches from 68 H&E images plus data augmentation to train the model for breast cancer in the original study. It is almost impossible to apply ST-Net to the typical Visium datasets with 1 section or 2 replicates. Moreover, in our experiments, we found out that the quality of their H&E images of the 12 human brain DLPFC replicates from the spatialLIBD project, is significantly worse, which results in poor performance for the top gene expression prediction.

For XFuse,

1. We first tried to follow the same imputation of a hold-out set in the experiment of the original work for possible comparison. However, the hold-out evaluation was to validate the imputation performance on only 100 highly variable genes for the spots in the hold-out region with Pearson correlation (Figure 1g). This evaluation would not be suitable for applying GNTD since such a small set of genes would not benefit from the connections in the whole PPI network.
2. Another challenge in practice is the expected running time. In our testing, XFuse was not scalable on the Stereo-seq datasets. The documentation of XFuse package also specifies that it will take more than three days to train the model on a standard Visium dataset with around 5k spots and 10k genes on a GPU card with 12GB GPU memory (There are runtime issues also reported by other users in the GitHub repository). This poses significant difficulty in fitting XFuse in cross-validation on the datasets for a reliable comparison.
3. Finally, we also expect other obstacles to compare XFuse with GNTD. By default, XFuse imputes a 2k-by-2k super-resolution expression map. The big size discrepancy to the standard 78x64 size of Visium array leads to doubt about applying XFuse in such a setting. While it is possible to introduce additional processing of super-resolution down to the actual size, XFuse does not model the reconstruction loss explicitly and it is unlikely to perform well in imputation by the evaluation metrics.

In summary, instead of making comparisons that are likely inconclusive, we believe it is better to conceive another study for developing a variation of GNTD to incorporate image features for imputation and a comprehensive evaluation of the performance in comparison with the other methods such as ST-net and XFuse. We added the discussion of XFuse and ST-Net in Section 4. Discussions and also mentioned possible ideas in future work in Section 5. Conclusions.

Their analyses did not take into account the resolution of spatial measurement, as noted in PMID: 35260632. While the figures in the manuscript suggest that predicted values fill the entire slide, this may be a misunderstanding for readers. If I understand correctly, their method cannot fill in unmeasured regions among measured spots. This should be mentioned as a limitation in the main text.

Responses: Thanks for pointing out the potential confusion in the previous submission. This revision includes more background review in Section 1. Introduction and discussion of the applicability and limitations of the current work in Section 4. Discussions, which clearly state the context of the imputation tasks and discuss image augmented imputation for improving the resolution and deconvolution with scRANseq data.

It was not clear how the authors selected their training, validation, and testing datasets. Then, I'm not sure these are fair comparisons (although the author said "for a fair comparison"). The authors should provide more details or a flowchart to clarify these procedures.

Responses: Thanks for the suggestion. We added a new Section 3.4 to explain the design of spot- and gene-wise 10-fold cross-validation for imputation evaluation. We also clarified the two training strategies for AE-based methods (i.e. training with all entries or non-zero entries only) in Section 3.7.

From the authors' codes, they used Pytorch for their analyses; however, there is no information in the main manuscript. At least version information is required.

Response: Thanks for the suggestion. We added a new Section 3.8 to summarize the Python environment and hardware specifications.

- **Line 440:** Although they used publicly available 10x Visium datasets from the manufacturer's website; however identification is very difficult. I recommend showing more detailed information, such as URL, unique name, or DOI.

Response: Thanks for the suggestion. We added a new Table S1 to summarize the datasets used in the study along with the data source and some meta information.

- **Line 454:** I suppose, before log transformation, the authors added pseudo-count values. There are several ways to normalize the UMI counts by library size. Please show exactly.

Response: Thanks for pointing out the unclear description of the normalization in the previous submission. We actually first used the Python package Scanpy to apply CPM normalization to raw UMI counts and then performed log transformation on the normalized expression after adding offset 1. The description of normalization is updated in lines 534-536.

- **Line 470:** Why top 15 PCs? Empirical selection in line 602 was also not reasonable. Determining from an elbow plot or some rules (i.e., total explained variance > 95%) is better. Based on lines 608-609, I felt that using 15 PCs is few.

Response: Thanks for raising the important question. The imputed data, especially those after non-linear embedding or factorization often already live on a low-dimensional space. For example, with the first two components by TSNE or UMAP, most of the clusters are already distinguishable visually. Thus, we do expect a relatively low number of PCs for optimal clustering with the imputed data by non-linear methods. We selected the top 15 PCs based on our empirical results and found out it provides the overall best performance on both Visium and Stereo-seq data across all the methods. The same number of PCs is used in the previous works BayesSpace and STAGATE, and we achieved very similar clustering performance to BayesSpace on raw data with 15 PCs. When more than 15 PCs are used, it does not lead to noticeable improvements but significantly more running time and memory usage of mclust. We also double-checked the explained variance by the first 15 PCs for all the imputed data. The captured variances are all above 85%.

- **Line 477:** Please explain BioGrid. In my understanding, STRING is a major database for PPI.

Response: Thanks for the comment. We did consider both BioGrid and STRING, which are both widely used repositories for PPIs. STRING might be more comprehensive by the inclusion of not only physical interactions but also interactions from text-mining, co-expression, and databases. In STRING, each interaction is also associated with a confidence score. The physical interactions from the BioGrid are actually a subset of interactions in the experimental channel from STRING. We used STRING for the experiment in our previous studies with different confidence cutoffs. In our experiments, the more stringent cutoff selecting only the physical interactions worked out the best. This is likely related to the type of interactions to consider for different types of data. We included an explanation in lines 561-563 to clarify this point.

- **Line 595:** What were the criteria for the early stopping?

Response: Thanks for the comment. We trained the GNTD on 90% non-zero entries and monitored the MSE on the remaining 10% non-zero entries, after running 1000 epochs, then early-stopped the training if the MSE had not improved within 50 epochs. We updated the description for early-stopping in lines 701-704.

- **Line 606:** The parameter for k-means clustering was not provided.

Response: Thanks for the comment. We used $k=100$ to cluster around 10k genes on mouse data and 20k genes on the human data, we also added the parameter of k-means for gene clustering in lines 735-736.

- **Line 620:** MSE and RMSE return almost the same information. Why did the authors use both?

Response: Thanks for pointing out our oversight. We removed the MSE metric from Equation 11.

Minor comments:

- **Line 645:** Need version information for MSigDB.

Response: Thanks for the comment, we added the version number of MSigDB in line 773.

- **Lines 652 – 653:** Need version information for these methods.

Response: Thanks for the comment. We added the version number of baseline models in lines 781-783.

Reviewer #3 (Remarks to the Author):

This manuscript presents a graph-guided neural tensor decomposition model named GNTD for reconstructing spatial transcriptomes (ST). GNTD first models the spatial coordinates and gene expressions with a 3-way tensor, then learns nonlinear latent factors to reconstruct the tensor via a multiple layer neural network. Comprehensive assessments on simulated and experimental datasets were conducted to show that GNTD outperformed existing methods in imputing ST data. The authors also demonstrated the effects of data imputation on downstream analysis tasks such as clustering.

1. GNTD adopts tensor decomposition for ST data imputation. However, its novelty is not clear as compared to the previously published FIST method (Li et al.). The frameworks of GNTD and FISH seem to be similar. It is thus important to highlight the differences between GNTD and FIST.

Response: Thanks for the comments. GNTD formulation is a graph-regularized hierarchical CPD while FIST formulation is a graph-regularized standard CPD. GNTD optimization is based on neural training while FIST's is gradient descent based on multiplicative updates. We added new paragraphs in Section 3.3 in the Methods to motivate GNTD by a hierarchical tensor decomposition framework, and we added more background review in Section 1. Introduction to compare GNTD and FIST.

2. In addition to SEDR and STAGATE, several more recent deep learning models such as GraphST (Long et al.) and SpaceFlow (Ren et al.) also output reconstructed spatial transcriptomics. Including them in the comparison would be informative to the readers.

Response: Many thanks for suggesting the recent works that can be potentially applied for spatial transcriptomics imputation. In the revision, GraphST has been included in the comparisons in Section 2.3 - 2.6, and The results are reported in imputation cross-validation in Figure 3, spot clustering in Figure 6, gene enrichment analysis in Figure 7, and experiments on stereo-seq data in Figure 8. SpaceFlow is built upon Deep Graph InfoMax (DGI), which trains an encoder to learn spot/cell embeddings to maximize local-global mutual information to capture expression patterns of spots/cells and their neighborhood. While SpaceFlow preserves spatial patterns in spots/cells embeddings, the model does not model the reconstruction loss explicitly or provide a decoder. Thus, the model does not output reconstructed expression like other AutoEncoder-based models (e.g. SEDR, STAGATE, and GraphST) for the comparison.

3. In Figure 3, the spatial expression patterns of three example genes were shown. It would be better to indicate the names of the three genes in the figure. How were these three genes selected? Or they are simply three random genes.

Response: Thanks for the comment and we apologize for the missing annotation in Figure 2. All these spatially variable genes were sampled based on the annotation on one of the human brain DFPLC sections. The three simulated genes are typical spatial variable genes enriching more than 2 out of the 7 annotated regions in the ground truth. In the revision, we showed the expression of all the 50 simulated spatially variable genes before and after the imputation in the supplementary Figure S3.

4. Most probe-based spatial transcriptomics technologies do not cover the whole transcriptome. For example, the cosmx technology by Nanostring is only able to probe 960 genes. Is it possible for GNTD to extend the 960 genes to the whole transcriptome if scRNA-seq data from the matching tissue type are available?

Response: Thanks for the interesting question. In the experiments, we evaluated both spot-wise and gene-wise predictions. While the spot-wise evaluation holds out the expressions of the entire spots in the test set, the gene-wise evaluation is performed by cross-validation within the expression values of the gene. Thus, it is different from the scenario of predicting the expressions of unprofiled genes using low-throughput spatial gene expression data that only profile less than a thousand genes. Potentially, GNTD can benefit from the connectivity in the PPI network for imputing the expressions of the unprofiled genes. However, the imputation of an unprofiled gene heavily depends on the number of neighboring genes that have been profiled, which will clearly limit the prediction to a small fraction of the unprofiled genes rather than the whole transcriptome. In addition, the small set of profiled genes might not

represent the full spatial expression landscape, and thus provide insufficient information for reconstructing the whole transcriptome. We postulate that it is better to integrate other kinds of data such as single-cell RNAseq data for such imputation task. GNTD formulation could be modified to share gene factors between spatial transcriptomics and scRNAseq data by a joint nonlinear decomposition. We discussed the future work in Section 4. Discussions.

5. Some imputation methods such as Tangram achieve ST reconstruction by integrating a reference scRNA-seq data with ST. I wonder whether reference-based methods like Tangram or reference-free methods like GNTD would perform better.

Response: Thanks for suggesting a comparison between GNTD and reference-based imputation methods such as Tangram. Since the primary goal of these methods is deconvolution rather than precise imputation, the nature of the imputed data is often entirely different from the ST expression data. For example, Tangram has a very different loss function, which searches the mapping between cells in single-cell transcriptomics and spots in spatial transcriptomics by maximizing the similarity between mapped single-cell expressions and spatial expression on a set of marker genes. To compare GNTD and Tangram, we performed the same 10-fold cross-validation on the region of the mouse brain cortex with matched scRNAseq data tested in the original study introducing Tangram. Then we measured both MAPE and R^2 for both methods and found out that Tangram performed much worse. This is expected since the imputation by Tangram is essentially aggregated from the single-cell expressions showing high correspondence to spatial expression, the single-cell transcriptomics might not agree with the spatial transcriptomics very well. We discussed this comparison in Section 4. Discussions. Note that we did not show the results since the MAPE and the R^2 by Tangram are much worse than the compared methods, and it is better to leave this topic for a more comprehensive study in our future work.

6. The data derived from spatial transcriptomics especially those with single-cell or subcellular resolution are believed to be sparse. A plot showing the sparsity level of the tested data before and after imputation with different methods would be helpful. More importantly, it has been observed that some tissue regions are more difficult to penetrate, and thus have higher sparsity level. It would be helpful to visualize the spatial distribution of sparsity before and after imputation.

Response: Thanks for the suggestions. In this revision, we included the visualization of the imputed data for each spatial variable gene in the simulation in Figure S3. We also added the ISH images of the marker gene expressions in Figure 9 for comparison with the imputed gene expressions. These results suggest that overall It is clear that the zeros and non-zeros are mostly well identified in the imputation by GNTD while discrepancy could exist in very sparsely expressed genes,

In terms of tissue penetration depth, we investigated two tissues human heart and human lymph node that might suffer from the penetration issue (Figure S10). We first identified regions overlapped with tissue but showing low density in expression, then we examined the correlation between the spot within the low-density region and its adjacent spot in the high-density region before and after imputation, it turns out that the correlation was significantly improved after imputation, which indicates GNTD somewhat addresses the penetration issue by introducing spot relations among spots.

7. Some imputation can distort the gene expression distribution. Pre- and post-imputation distribution plots are needed to confirm that the imputation does not introduce data distortion.

Response: Thanks for the comment. We added a plot of expression distributions of pre- and post-imputation data on the 22 Visium and 3 StereoSeq datasets in Supplementary Figure S5. The expressions are quite similar without significant distortion on the Visium data. The distribution of the raw expressions on the StereoSeq data shows some bi-modal/multi-modal patterns, but the means of the components are quite similar. Thus, the imputation with one peak can still be a good approximation. We also checked the imputed data by other methods such as DTD and CosTCO. They show similar distributions with no surprise since all the compared methods are based on MSE-like loss.

8. Figure 7 shows the results of enrichment analysis of spatially co-expressed gene clusters. If possible, it would be helpful to add a correlation heatmap plot to show the spatial co-expression patterns of the gene clusters identified before and after imputation.

Response: Thanks for the suggestion on improving the interpretability of the results in Figure 7. Since visualizing the clustering of tens of thousands of genes in a heatmap is difficult, we show the mean expression patterns of some clusterings in Figure S7 and report the detailed gene enrichment analysis in Table S2.

9. In Figure 9B, the last three genes showed nearly zero expressions before imputation. However, after imputation their expressions became very high and detected all over the tissue. Such substantial changes need to be validated, for example with RNAscope or multiplex IHC/IF to confirm their expressions.

Response: Thanks for pointing out the visualization and clarity issue in Figure 9B. The reason for such visualization is because each gene plot in Figure 9B has its own color scale based on its minimal and maximal expression for better visualization of spatial patterns. The values are not directly comparable across the methods. GNTD imputation actually does not significantly change the expression levels for non-zero entries since the MSE loss is between the raw and imputed data for all non-zero entries. In the revision, we showed marker gene visualization with the same color scale for both DLPFC 151673 section from Visium and mouse olfactory bulb from Stereo-seq in the supplementary Figure S8. It is clear that the imputed expressions are all in the scale as the raw data. In addition, we also included the ISH images of the marker gene expressions in Figure 9 for comparison with the imputed gene expressions.

10. An evaluation of memory usage and runtime with respect to data size can guide users to choose the best method for their data analysis task.

Response: Thanks for the suggestion. We added a new Section 3.8 to provide the information on implementation, running environment, and running time on Visium data and Stereo-seq data.

11. When testing SEDR, the authors built the spatial graph among spots by combining spot neighbourhood and co-expression information. However, in the original SEDR model, only spot neighbourhood is used to build the graph. Such inconsistency may lead to poorer performance of SEDR.

Response: We apologize for the confusion. In fact, we only used both co-expression and spatial localization to build the graph for GNTD, and did not involve co-expressions in constructing the spatial graph for SEDR. We clarified this experiment setup in Section 3.7.

12. To evaluate the imputation performance, the authors compared five methods, including three tensor-based models CoSTCo, DTD, and FIST, and two deep learning models SEDR, and STAGATE. But only the tensor-based models were compared with GNTD on the simulated data. The authors should justify the reasons.

Response: Thanks for the comment. Our intention of the simulation was to evaluate tensor imputation only and thus didn't compare the deep learning models. In this revision, we also included SEDR, STAGATE, and GraphST in the comparison on the simulated data. The results are shown in Figure S1.

13. Figures 3 and 8 show negative values for both SEDR and STAGATE in terms of the R2 metric. Further explanations are required to clarify this.

Response: Thanks for the question on the R^2 metric. Note that the R^2 metric as defined in Equation 11 can be negative when the overall prediction is worse than the mean. This can happen very often in sparse data if the non-zero entries are not correctly predicted from the majority of zeros. In addition, the AE-based models (STAGATE, SEDR, and GraphST) are mainly designed to learn embeddings, the prediction by these models with non-zero training predicted expression not even following the original spatial pattern in the raw data. This is evident in our simulation results in Figure S1. We added the explanation in Section 2.3.

14. In the clustering analysis, some results look a bit weird, such as the results of CoSTCo, STAGATGE, and FIST on slice 151673 (Figure 6), and results of STAGATE on mouse brain Stereo-seq data. The authors should double-check their codes and ensure the accuracy of all results presented in the manuscript.

Response: Thanks for identifying some issues in the comparison in Figure 6. There was no implementation issue in the experiment. The discrepancy is from the strategy for training the deep learning models. There are two kinds of training to learn the neural network models, 1) training with only non-zero entries or 2) all the entries (including zero entries), specified by the implementation of the loss function. Note that here all the entries are always used as input and the discrepancy is in which entries are penalized in the loss function. In the previous submission, we only showed the clustering results based on the imputation by using the non-zero training setting for STAGATE in Figure 6. In this revision, we trained the baseline methods, SEDR, STAGATE, and GraphST with both settings and further optimized the number of hidden units in the neural network. Using the new results by training with all entries, the results in Figure 6 are fully optimized for clustering with these methods. For the tensor completion models, DTD and CoSTCo, the loss is always measured on the non-zero entries only, similarly as they were originally designed for scalable sparse data imputation. CoSTCo and FIST might suffer from overfitting since the data from spatialLIBD are relatively more sparse compared to the other Visium data. We clarified these settings in Section 3.7.

15. In the method section, the symbols and functions need to be clearly defined. For example, symbols such as n_g , n_y , and n_x are undefined. What $f_m()$ and $f_{agg}()$ represent should be described.

Response: Thanks for the comment. We revised and checked the method in Section 3 to make sure that all symbols and functions are appropriately defined.

16. The authors claim that they trained deep learning methods SEDR and STAGATE with non-zero entries to do the imputation. However, both SEDR and STAGATE typically take the whole gene expressions as input features during model training by default. It should be noted that using non-zero entries for training these models may impact their performance since the default parameters assume whole gene expressions as inputs.

Response: Thanks again for the suggestion on improving the experimental setting for SEDR and GraphST. As we mentioned in the response to comment #14 above, we trained all the AE-based models, including SEDR, STAGATE, and newly added GraphST with all entries and reported the new results in Sections 2.4-2.6. We also optimized the hyperparameter (mainly the number of hidden units) for these AE-based models by examining their loss function on the simulated data, Visium, and Stereo-seq data to make sure the model is properly fitting to each dataset for the comparison.

Reviewer #1 (Remarks to the Author):

The authors have addressed all my comments.

Reviewer #2 (Remarks to the Author):

The authors have addressed almost all of my concerns. The followings are the remaining comments relating to my previous comments.

Line 877: The authors should show the details about "(results not shown)" in Supplementary Information.

Lines 894–898: In the revised version, the authors added these lines based on the responses to my first major comment. The responses were reasonable and well-written; however, they significantly simplified and shortened the original responses in these lines in the main text. For example, there are no details about their trials in the main text, resulting in making it difficult to understand their statement, "While these methods have been shown to perform well in imputing a small number of genes in the original studies, the applicability to the whole transcriptome seems to be rather limited in both the scalability and lack of a complete evaluation.". Maybe readers will think there is insufficient evidence to claim the XFuse and ST-Net methods. Then, I recommend adding full details of the responses to my first major comment in the Supplementary Text.

Lines 114–120: This very long sentence has strange English grammar and should be separated.

Reviewer #3 (Remarks to the Author):

The authors have addressed my questions. I have no further questions.

Revision Summary

We sincerely thank all the reviewers for their constructive comments, which have led to significant improvement and more insight of the research work in this manuscript. In this revision, we addressed the comments from **Review 2** as follows,

(1) Line 877: The authors should show the details about “(results not shown)” in Supplementary Information.

(2) Lines 894–898: In the revised version, the authors added these lines based on the responses to my first major comment. The responses were reasonable and well-written; however, they significantly simplified and shortened the original responses in these lines in the main text. For example, there are no details about their trials in the main text, resulting in making it difficult to understand their statement, “While these methods have been shown to perform well in imputing a small number of genes in the original studies, the applicability to the whole transcriptome seems to be rather limited in both the scalability and lack of a complete evaluation.”. Maybe readers will think there is insufficient evidence to claim the XFuse and ST-Net methods. Then, I recommend adding full details of the responses to my first major comment in the Supplementary Text.

Response: Thanks for the suggestion. We have added a new section “Comparisons to reference-based imputation methods” in the supplementary document to provide the detailed discussions of Xfuse, ST-Net and the results of Tangram.

(3) Lines 114–120: This very long sentence has strange English grammar and should be separated.

Response: This paragraph has been rewritten to correct the grammar issue.

We highlighted all the revised text in the manuscript.